# Genetic and neuronal regulation of sleep by neuropeptide VF

**Daniel A Lee[1], Andrey Andreev[2], Thai V Truong[3], Audrey Chen[1], Andrew J Hill[1], Grigorios Oikonomou[1], Uyen Pham[1], Young K Hong[1], Steven Tran[1], Laura Glass[1], Viveca Sapin[1], Jae Engle[1], Scott E Fraser[2,3], David A Prober[1]***

[1]Division of Biology and Biological Engineering, California Institute of Technology, Pasadena, United States; [2]Department of Bioengineering, University of Southern California, Los Angeles, United States; [3]Translational Imaging Center, University of Southern California, Los Angeles, United States

**Abstract** Sleep is an essential and phylogenetically conserved behavioral state, but it remains unclear to what extent genes identified in invertebrates also regulate vertebrate sleep. RFamide-related neuropeptides have been shown to promote invertebrate sleep, and here we report that the vertebrate hypothalamic RFamide neuropeptide VF (NPVF) regulates sleep in the zebrafish, a diurnal vertebrate. We found that NPVF signaling and *npvf*-expressing neurons are both necessary and sufficient to promote sleep, that mature peptides derived from the NPVF preproprotein promote sleep in a synergistic manner, and that stimulation of *npvf*-expressing neurons induces neuronal activity levels consistent with normal sleep. These results identify NPVF signaling and *npvf*-expressing neurons as a novel vertebrate sleep-promoting system and suggest that RFamide neuropeptides participate in an ancient and central aspect of sleep control.

DOI: https://doi.org/10.7554/eLife.25727.001

## Introduction

Sleep is an essential, phylogenetically conserved behavioral state whose regulation requires several brain regions that engage multiple neurochemical systems (*Allada and Siegel, 2008*; *Cirelli, 2009*). Historically, stereotactic lesion experiments have demonstrated a central role in sleep regulation for the hypothalamus (*Saper et al., 2010*; *Weber and Dan, 2016*), a structure that is anatomically and molecularly conserved between zebrafish and mammals (*Biran et al., 2015*; *Chiu and Prober, 2013*; *Liu et al., 2015*; *Shimogori et al., 2010*). More recent genetic approaches have corroborated a critical role for hypothalamic regulation of vertebrate sleep, due in part to production of specific wake- and sleep-promoting neuropeptides (*Richter et al., 2014*). However, only a small number of such neuropeptides have been identified, suggesting that additional neuropeptides remain to be discovered. Indeed, the hypothalamus produces many neuropeptides (*van den Pol, 2012*), but potential roles in sleep for most of these neuropeptides, and the neurons that produce them, are largely unknown.

Sleep has primarily been studied using mammals, where electrophysiology can be used to define sleep and wake states. However, progress has been challenging, due in part to the complexity of mammalian brains, the significant cost and labor required to generate and test mutant and transgenic rodents, and the poor amenability of mammalian model systems for large-scale screens. To overcome these limitations, several groups have established invertebrate sleep models by using behavioral criteria to define sleep-like states (*Allada and Siegel, 2008*; *Cirelli, 2009*). First, sleep primarily occurs during specific periods of the circadian cycle during which animals usually adopt an inactive and specific posture. In mice, an inactive period of $\geq 40$ seconds significantly correlates with electroencephalographic indications of sleep (*Pack et al., 2007*). Second, animals exhibit an

**\*For correspondence:** dprober@caltech.edu

**Competing interests:** The authors declare that no competing interests exist.

increased arousal threshold during sleep, thus distinguishing sleep from quiet wakefulness. However, sleeping animals can still be aroused by strong stimuli, which distinguishes sleep from paralysis or coma. Third, sleep is controlled by a homeostatic system, which can be demonstrated by an increased need for sleep following sleep deprivation. Based on these criteria, rest in nematodes, flies and zebrafish (*Hendricks et al., 2000*; *Prober et al., 2006*; *Raizen et al., 2008*; *Shaw et al., 2000*; *Van Buskirk and Sternberg, 2007*; *Yokogawa et al., 2007*; *Zhdanova et al., 2001*) has been shown to be a sleep-like state (for simplicity, hereafter referred to as sleep). These model organisms have been used to perform genetic and genomic screens that have identified many genes that affect sleep (*Chiu et al., 2016*; *Cirelli et al., 2005*; *Koh et al., 2008*; *Nath et al., 2016*; *Shang et al., 2013*; *Shi et al., 2014*; *Trojanowski et al., 2015*), including several neuropeptides. However, hypothalamic neuropeptides implicated in regulating vertebrate sleep, including hypocretin (Hcrt) (*Adamantidis et al., 2007*; *de Lecea et al., 1998*; *Lin et al., 1999*) and melanin concentrating hormone (MCH) (*Konadhode et al., 2013*; *Tsunematsu et al., 2014*; *Verret et al., 2003*; *Willie et al., 2008*) lack clear invertebrate orthologs, suggesting that neuropeptidergic regulation of sleep may have evolved independently in vertebrates and invertebrates. Thus, it remains to be determined whether neuropeptides that regulate invertebrate sleep are functionally conserved in vertebrates.

Members of a neuropeptide family that contain a C-terminal Arg-Phe-$NH_2$ motif, known as RFamides, play key roles in promoting invertebrate sleep (*He et al., 2013*; *Iannacone et al., 2017*; *Lenz et al., 2015*; *Nagy et al., 2014*; *Nath et al., 2016*; *Nelson et al., 2014*; *Shang et al., 2013*; *Turek et al., 2016*). Vertebrates possess several RFamide family members (*Elphick and Mirabeau, 2014*), including the hypothalamic neuropeptide VF (NPVF). Mammalian NPVF (also known as RFRP or GnIH) was first identified in a bioinformatic screen for human RFamide neuropeptides (*Hinuma et al., 2000*). The human NPVF preproprotein produces three mature peptides, known as RFamide-related peptide (RFRP) 1, RFRP2, and RFRP3 (*Hinuma et al., 2000*). In rodents, intracerebroventricular infusion of synthetic RFRP1 or RFRP3 has been reported to affect hormone levels, food intake, sexual behavior and nociception (*Kim, 2016*; *Kriegsfeld et al., 2015*; *Liu et al., 2001*; *Tsutsui et al., 2015*; *Ubuka et al., 2012*). RFRP1 and RFRP3 can bind to NPFFR1 and NPFFR2 in mammalian cell culture, but RFRP2 does not and its cognate receptor(s) remains unknown (*Hinuma et al., 2000*; *Liu et al., 2001*). In the brain, NPVF is enriched within and around the dorsomedial hypothalamus in rodents (*Liu et al., 2001*; *Poling et al., 2012*) and humans (*Ubuka et al., 2009*), a region associated with sleep regulation (*Saper et al., 2010*). However, genetic gain- or loss-of-function sleep studies of mammalian NPVF-derived neuropeptides or *npvf*-expressing neurons (hereafter referred to as NPVF neurons) have not been reported.

Here we address the role of NPVF in regulating vertebrate sleep using the zebrafish, a vertebrate that allows for rapid and cost-effective testing of genes for effects on behavior (*Chiu et al., 2016*). The zebrafish has been shown to exhibit a diurnal pattern of sleep/wake states using the behavioral criteria described above. Based on locomotor activity and arousal threshold assays, one or more minutes of inactivity correspond to a sleep state in zebrafish larvae (*Elbaz et al., 2012*; *Prober et al., 2006*). The zebrafish has genetic (*Cirelli, 2009*; *Richter et al., 2014*; *Sehgal and Mignot, 2011*), pharmacological (*MacRae and Peterson, 2015*) and neuroanatomical (*Kaslin and Panula, 2001*; *McLean and Fetcho, 2004*; *Prober et al., 2006*) similarities to mammalian sleep, suggesting that findings in zebrafish will translate to mammals. Using genetics, pharmacology, chemogenetics and optogenetics, we show that NPVF signaling and NPVF neurons are both necessary and sufficient to promote sleep, and that stimulation of NPVF neurons induces levels of neuronal activity consistent with normal sleep. These results establish RFamide neuropeptides as evolutionarily conserved sleep-promoting systems.

## Results

### Overexpression of NPVF promotes sleep in zebrafish larvae

Recent studies using *Drosophila melanogaster* (*Drosophila*) and *Caenorhabditis elegans* (*C. elegans*) demonstrated that several RFamide peptides, as well as the neurons that produce them, are necessary and sufficient to promote invertebrate sleep (*He et al., 2013*; *Iannacone et al., 2017*; *Lenz et al., 2015*; *Nath et al., 2016*; *Nelson et al., 2014*; *Shang et al., 2013*; *Trojanowski et al., 2015*; *Turek et al., 2016*). A library of peptides was previously used to screen for zebrafish sleep

phenotypes, but the library did not include RFamide peptides (*Chiu et al., 2016*). We therefore set out to test the hypothesis that RFamide peptides can also promote sleep in vertebrates. One *C. elegans* sleep-promoting RFamide peptide, FLP-13, is sufficient to induce sleep (*Nath et al., 2016*; *Nelson et al., 2014*) and is necessary for sleep in response to cellular stress (*Nelson et al., 2014*). The vertebrate RFamide whose mature peptide sequences most closely resemble those of FLP-13 is NPVF. The three mature peptides derived from NPVF (RFRP1-3) contain a conserved C-terminal

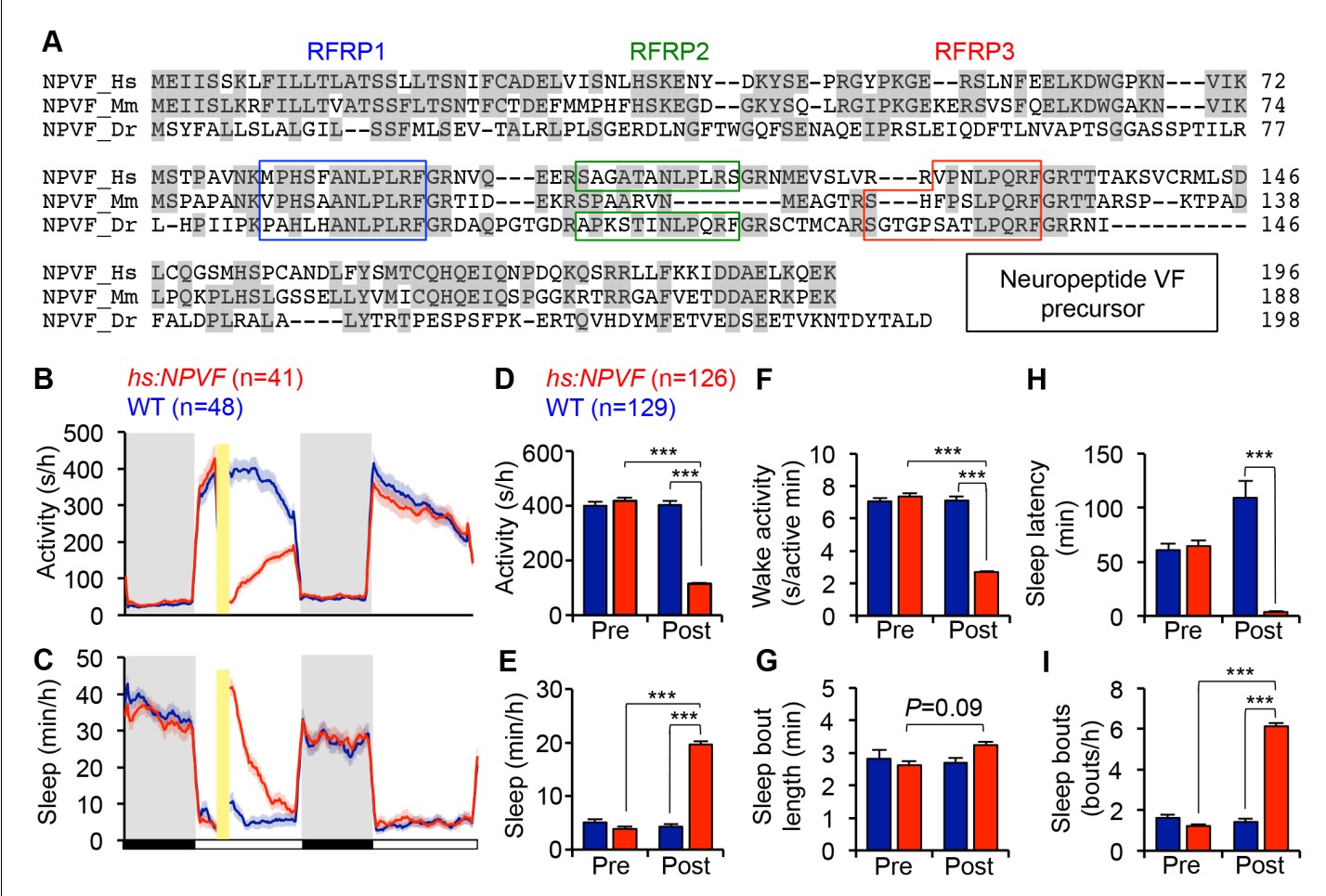

**Figure 1.** NPVF overexpression decreases locomotor activity and increases sleep. (A) Multiple sequence alignment of human (Hs), mouse (Mm), and zebrafish (Dr) NPVF preproproteins. Colored boxes demarcate identified or predicted mature peptide sequences. Note that the mouse NPVF protein lacks RFRP2. (B–I) NPVF overexpression decreased locomotor activity (B,D) and increased sleep (C,E) in transgenic animals compared to WT siblings and to pre-HS. Yellow bar indicates heat shock. Pre- and Post-HS data is calculated for the day of HS. White and black bars under behavioral traces indicate day (14 h) and night (10 h), respectively. NPVF overexpression decreased wake activity (F), decreased sleep latency (time to first sleep bout following lights on in the morning Pre-HS, or following HS in post-HS measurement) (H) and increased sleep bout number (I), and also caused a trend of increased sleep bout length (G). Mean ± SEM from one representative experiment (B,C), or three pooled experiments (D–I) are shown. n = number of animals. ***p<0.0001 by Two-way ANOVA with Holm-Sidak test. See also *Figure 1—figure supplements 1–3*.

DOI: https://doi.org/10.7554/eLife.25727.002

The following figure supplements are available for figure 1:

**Figure supplement 1.** Overexpression of *C. elegans* FLP-13 decreases locomotor activity in zebrafish larvae.
DOI: https://doi.org/10.7554/eLife.25727.003

**Figure supplement 2.** NPVF overexpression in the evening promotes sleep the following night and day.
DOI: https://doi.org/10.7554/eLife.25727.004

**Figure supplement 3.** *npvf* expression levels do not vary in a circadian manner and NPVF overexpression-induced sleep does not require entrained behavioral circadian rhythms.
DOI: https://doi.org/10.7554/eLife.25727.005

LPXRF motif (where X = L or Q) (*Figure 1A*) that is critical for peptide function (*Findeisen et al., 2011*). The nine peptides derived from FLP-13 contain a similar C-terminal PXIRF motif (where X = L or F) (*Figure 1—figure supplement 1A*).

To determine whether NPVF overexpression affects sleep, we generated a stable transgenic zebrafish line in which a heat shock-inducible promoter regulates expression of the zebrafish *npvf* open reading frame. After heat shock-induced overexpression of NPVF, we found that these animals were 71% less active, had 62% less wake activity (amount of activity while awake), and slept 350% more during the day compared to their non-transgenic wild-type (WT) siblings (*Figure 1B–1F*) (activity, wake activity and sleep: p<0.0001, Two-way ANOVA, Holm-Sidak test). The increase in sleep was primarily due to an increase in the number of sleep bouts (+331%, p<0.0001, Two-way ANOVA, Holm-Sidak test), along with a modest increase in the length of sleep bouts (+21%) that did not reach statistical significance (*Figure 1G and I*). NPVF overexpression also drove the transition from wake-to-sleep by 2850% compared to WT siblings, with the average transgenic animal falling asleep in 3.7 ± 0.4 min immediately following heat shock compared to 108.9 ± 16.0 min for WT siblings (p<0.0001, Two-way ANOVA, Holm-Sidak test, *Figure 1H*).

Interestingly, induction of NPVF overexpression in the middle of the day did not affect sleep the following night. A previous study suggested that the circadian system prevents premature sleep onset in the evening, when homeostatic sleep drive is high (*Cavanaugh et al., 2016*). However, when we induced NPVF overexpression at the end of the day, fifteen minutes before lights off, we observed significantly decreased locomotor activity (−36%) and increased sleep (+20%) during the night, as well as the following day (Locomotor activity: −14%; Sleep: 124%) compared to WT siblings (all: p<0.0001, Two-way ANOVA, Holm-Sidak test, *Figure 1—figure supplement 2*), although the effect during the following day was smaller than that observed when NPVF overexpression was induced in the middle of the day. These observations suggest that the absence of a phenotype at night following heat shock during the middle of the day is due to declining levels of overexpressed NPVF protein, rather than circadian regulation of NPVF function. This interpretation is supported by our observation that the level of *npvf* mRNA, assayed using RT-qPCR and *in situ* hybridization (ISH), does not vary in a circadian manner (p>0.05, One-way ANOVA, *Figure 1—figure supplement 3A and B*).

To further test whether NPVF-induced sleep is regulated by circadian cues, we raised and tested animals in either constant light (LL) or dark (DD) from birth, which prevents the establishment of circadian rhythms (*Gandhi et al., 2015*; *Hurd and Cahill, 2002*; *Kaneko and Cahill, 2005*). Following heat-shock in LL, NPVF-overexpressing animals were 60% less active and slept 75% more than their WT siblings (*Figure 1—figure supplement 3C–G*) (activity and sleep: p<0.0001, Two-way ANOVA, Holm-Sidak test). To test whether light affects this phenotype, we raised and tested animals in DD, and observed similar effects on overall activity and sleep as in LL (*Figure 1—figure supplement 3J–N*). Interestingly, we observed differences in sleep architecture between LL and DD. In LL, the increase in NPVF-induced sleep was due to a 200% increase in the number of sleep bouts compared to WT siblings (p<0.0001; Two-way ANOVA, Holm-Sidak test) (*Figure 1—figure supplement 3H and I*). In DD, increased sleep in NPVF-overexpressing animals was due to a 75% increase in sleep bout length compared to WT siblings (p<0.0001, Two-way ANOVA, Holm-Sidak test), with no effect on the number of sleep bouts (*Figure 1—figure supplement 3O and P*). Together, these results suggest that NPVF-induced sleep does not require circadian rhythms or depend on lighting conditions, although effects on sleep architecture are influenced by lighting conditions.

These results indicate that the ability of RFamide neuropeptides to promote sleep is conserved in invertebrates (*He et al., 2013*; *Iannacone et al., 2017*; *Lenz et al., 2015*; *Nath et al., 2016*; *Nelson et al., 2014*; *Shang et al., 2013*; *Trojanowski et al., 2015*; *Turek et al., 2016*) and vertebrates. We next asked whether the *C. elegans* sleep-promoting RFamide FLP-13 is sufficient to affect behavior in zebrafish. To answer this question, we generated transgenic zebrafish in which expression of the *C. elegans flp-13* open reading frame is regulated by a heat shock-inducible promoter. We found that overexpression of FLP-13 decreased locomotor activity by 26% compared to WT siblings (*Figure 1—figure supplement 1B*) (p<0.0001, Two-way ANOVA, Holm-Sidak test). However, FLP-13 overexpression did not significantly affect sleep (*Figure 1—figure supplement 1C*). These results are consistent with the notion that RFamide peptides have an evolutionarily conserved role in suppressing motor behaviors, even though the behavioral effect of overexpressing FLP-13 in zebrafish is weaker than that of its potential zebrafish homolog NPVF.

## Mature peptides derived from NPVF synergistically promote sleep

In humans and zebrafish, *npvf* encodes for a preproprotein that contains three mature peptides (RFRP1-3) (*Ubuka and Tsutsui, 2014*). To determine whether individual mature peptides are sufficient to promote sleep, we generated transgenic zebrafish that contained heat-shock inducible transgenes in which the amino acid sequence of one or more mature peptides was scrambled to abolish its function (*Figure 2*). Heterozygous transgenic animals were mated to WT animals, and their heterozygous transgenic and WT larval progeny were assayed for locomotor activity and sleep phenotypes. Overexpression of a transgene in which all three peptides were scrambled had no behavioral effect (*Figure 2D–F and N*), indicating that one or more peptides are required for NPVF overexpression-induced sleep. Compared to their WT siblings, overexpression of transgenes containing only RFRP1 or RFRP3 decreased activity by 18% or 14%, respectively (both: p<0.0001, One-way ANOVA, Holm-Sidak test), but had no significant effect on sleep (*Figure 2D, E, G, I, O and Q*). Overexpression of RFRP2 (present in the human but not rodent *npvf* ortholog) had a stronger effect, decreasing locomotor activity by 56% and increasing sleep by 120% (activity and sleep: p<0.0001, One-way ANOVA, Holm-Sidak test) (*Figure 2D, E, H and P*). Compared to their WT siblings, overexpression of transgenes containing any combination of two RFRP peptides induced significantly larger activity and sleep phenotypes, similar to overexpression of WT NPVF, with >70% decreased locomotor activity and >200% increased sleep (activity and sleep: p<0.0001, One-way ANOVA, Holm-Sidak test) (*Figure 2D, E, J–M and R–U*).

To test whether the behavioral effects induced by overexpression of two or more RFRP peptides is due to a dosage effect, we compared sleep in animals containing 0 (WT), 1 (heterozygous transgenic) or 2 (homozygous transgenic) copies of the *hs:RFRP1*, *hs:RFRP3* or *hs:RFRP1-3* transgenes. Following heat-shock induced overexpression, we observed no difference in locomotor activity or sleep amount between WT, *Tg(hs:RFRP1)/+* and *Tg(hs:RFRP1)/Tg(hs:RFRP1)* animals (*Figure 2—figure supplement 1A–D*). Similarly, we observed no difference in amount of locomotor activity or sleep between *Tg(hs:RFRP3)/+* and *Tg(hs:RFRP3)/Tg(hs:RFRP3)* animals (*Figure 2—figure supplement 1F and H*). We observed a small decrease in locomotor activity in *Tg(hs:RFRP3)/Tg(hs:RFRP3)* compared to +/+ animals (*Figure 2—figure supplement 1E and G*), but the effect was much smaller than that observed for *Tg(hs:RFRP1,3)/+* compared to +/+ animals (*Figure 2K and S*). Finally, we compared +/+, *Tg(hs:RFRP1-3)/+*, and *Tg(hs:RFRP1-3/hs:RFRP1-3)* animals and observed that RFRP1-3 overexpression increased sleep and decreased locomotor activity to a similar extent in heterozygous and homozygous transgenic animals compared to +/+ siblings (*Figure 2—figure supplement 1I–L*). Taken together, these results suggest that the larger phenotypes observed following overexpression of 2 or more different RFRP peptides compared to single peptides is due to the synergistic action of these peptides rather than a dosage effect.

To compare the effect of overexpressing different RFRP peptides on neuronal activity in the brain, we performed ISH using a probe specific for *c-fos*, which can be used as a marker of neuronal activity (*Guzowski et al., 2005*). One hour after heat shock, we observed robust induction of *c-fos* expression along the brain ventricular lining in transgenic animals that overexpressed WT *npvf*, whereas their non-transgenic siblings lacked *c-fos* expression in this region (*Figure 2—figure supplement 2A–C*). In several vertebrates, including fish, hypothalamic RFamidergic neurons have been described as cerebrospinal fluid (CSF)-contacting neurons (*Castro et al., 2001*), suggesting that activation of cells along the brain ventricle by RFRP peptides could drive behavioral state changes by inducing signals in the CSF to promote sleep, as has been described for several sleep regulators (*Krueger et al., 2007*). Consistent with the behavioral phenotypes induced by overexpression of different RFRP peptides, we observed robust ventricular *c-fos* expression in animals that overexpressed any combination of two RFRP peptides (*Figure 2—figure supplement 2J* and data not shown) or RFRP2 alone (*Figure 2—figure supplement 2H*), reduced *c-fos* expression in animals that overexpressed RFRP1 or RFRP3 (*Figure 2—figure supplement 2G and I*), and no *c-fos* expression in animals that overexpressed a transgene in which all three peptides were scrambled (*Figure 2—figure supplement 2F*). Taken together with the behavioral data (*Figure 2* and *Figure 2—figure supplement 1*), these changes in *c-fos* expression suggest that NPVF overexpression-induced sleep results from synergistic effects of RFRP peptides.

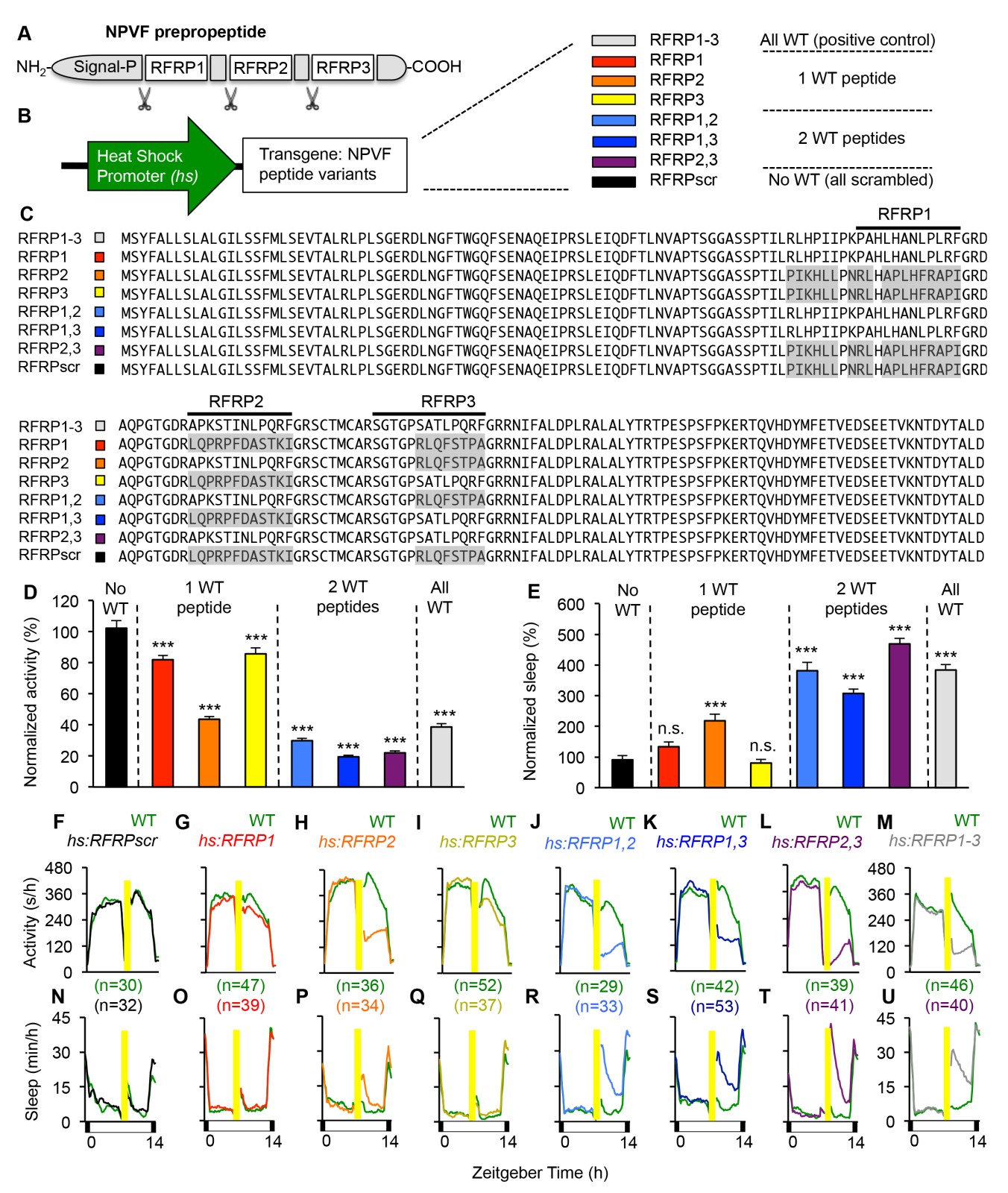

**Figure 2.** Individual RFRP peptides act synergistically to promote sleep. (**A**) Schematic diagram of the NPVF preproprotein and its three RFRP peptides. (**B**) Eight heat shock-inducible transgenes containing all possible combinations of one, two or three scrambled RFRP peptides tested. Colors correspond to those used in panels (**C–U**). (**C**) Amino acid sequences of zebrafish *Tg(hs:NPVF)* transgenes containing single or multiple scrambled RFRP peptides. RFRP peptide sequences are predicted based on conservation with peptides isolated in other species and on the location of glycine

*Figure 2 continued on next page*

*Figure 2 continued*

amidation signals and lysine or arginine endoproteolytic cleavage sites. The C-terminus of each predicted peptide contains a LPXRF motif, where X = L or Q. Gray shading indicates scrambled sequences. Daytime activity (D) and sleep (E) of transgenic animals normalized to their WT siblings following afternoon heat-shock-induced transgene overexpression are shown as mean ± SEM for two pooled experiments, with at least 64 animals per genotype. Examples of activity and sleep data for each transgene are shown in (F–U). Yellow bars indicate heat shock. White and black bars under behavioral traces indicate day (14 hr) and night, respectively. n = number of animals. n.s. = not significant, ***p<0.0001 by One-way ANOVA compared to scrambled NPVF peptide control (black) using Dunnett's multiple comparison test. See also *Figure 2—figure supplements 1* and *2*.
DOI: https://doi.org/10.7554/eLife.25727.006

The following figure supplements are available for figure 2:

**Figure supplement 1.** RFRP-induced sleep is not due to a dose-response of individual RFRP peptides.
DOI: https://doi.org/10.7554/eLife.25727.007
**Figure supplement 2.** *c-fos* expression correlates with overexpression of RFRP peptides that induce sleep.
DOI: https://doi.org/10.7554/eLife.25727.008

## Loss of NPVF signaling results in reduced sleep

We next tested the hypothesis that NPVF signaling is required for normal sleep levels. Using CRISPR/Cas9, we generated a zebrafish *npvf* mutant that contains a 7 bp deletion after the first amino acid of RFRP2 (*Figure 3—figure supplement 1A*). This mutation results in a shift in the translational reading frame and produces a protein that contains RFRP1 but lacks RFRP2 and RFRP3. Consistent with the NPVF overexpression phenotype, we found that *npvf* -/- animals were 27% more active and slept 10% less than their *npvf* +/+ siblings at night (sleep: p<0.05; activity: p<0.005, One-way ANOVA, Holm-Sidak Test) (*Figure 3A–D*). These effects were due to a 10% reduction in sleep bout length and a 19% increase in wake activity in *npvf* -/- animals compared to their *npvf* +/+ sibblings (p<0.01 and p<0.005, respectively, One-way ANOVA, Holm-Sidak test) (*Figure 3—figure supplement 1B–E*). *npvf* +/- animals exhibited an intermediate sleep phenotype that was not significantly different from that of their *npvf* -/- siblings (night: p=0.3, One-way ANOVA, Holm-Sidak test) (*Figure 3C and D*). This result suggests that NPVF signaling is required for normal sleep levels.

Because the behavioral phenotype of *npvf* -/- animals, which chronically lack NPVF signaling, is relatively mild, we used pharmacology as an alternative approach to acutely inhibit NPVF signaling. We tested two pharmacological inhibitors of NPFFR1 and NPFFR2, the GPCRs that bind RFRP1 and RFRP3 *in vitro* (*Hinuma et al., 2000*; *Liu et al., 2001*). Treatment of WT animals with the dipeptide antagonist RF9 (*Simonin et al., 2006*) resulted in more activity (day: 28%; night: 20%) and less sleep (day: −47%; night: −24%) than animals treated with DMSO vehicle control during both the day and night (p<0.0001, Student's *t*-test) (*Figure 3E–3H*). Changes in sleep were associated with fewer and shorter sleep bouts at night, but only to fewer sleep bouts during the day (*Figure 3—figure supplement 1G and H*). RF9 also significantly increased wake activity during the day (21%) and night (7%), and sleep latency after lights off at night (74%) compared to DMSO-treated controls (p<0.0001, Student's *t*-test) (*Figure 3—figure supplement 1F and I*). Similarly, treatment of WT animals with GJ-14, an additional dipeptide antagonist of NPFFR1 and NPFFR2 (*Kim et al., 2015*), resulted in significantly increased locomotor activity (day: 9%; night: 43%) and decreased sleep (day: −36%; night: −24%) during the day and night (p<0.0001, Student's *t*-test) (*Figure 3I–3L*). Although the effects of GJ-14 were smaller than those of RF9, GJ-14 had similar effects on sleep architecture as RF9 (*Figure 3—figure supplement 1K and L*). These results suggest that NPVF signaling is required for both the initiation and maintenance of normal levels of sleep during the day and night.

To confirm that RF9 and GJ-14 are acting by inhibiting NPFFR receptors, we tested whether these antagonists block sleep induced by overexpression of RFRP1,3, the cognate ligands of these receptors. In independent behavioral experiments, *Tg(hs:RFRP1,3)* animals and their WT siblings were treated with either RF9, GJ-14, or DMSO vehicle control in clutch matched controls. RFRP1,3 overexpression induced sleep post-HS compared to pre-HS in DMSO vehicle treated animals (*Figure 3—figure supplement 2A, B, E and F*). In contrast, RFRP1,3 overexpression-induced sleep was suppressed post-HS compared to pre-HS in animals treated with RF9 (*Figure 3—figure supplement 2C and D*) and completely blocked by GJ-14 treatment (*Figure 3—figure supplement 2G and H*). These results are consistent with *in vitro* data showing that RFRP1 and RFRP3 bind NPFFR1 and NPFFR2 (*Hinuma et al., 2000*; *Liu et al., 2001*). These data suggest that decreased sleep due to treatment with these drugs can be attributed to inhibition of NPVF signaling.

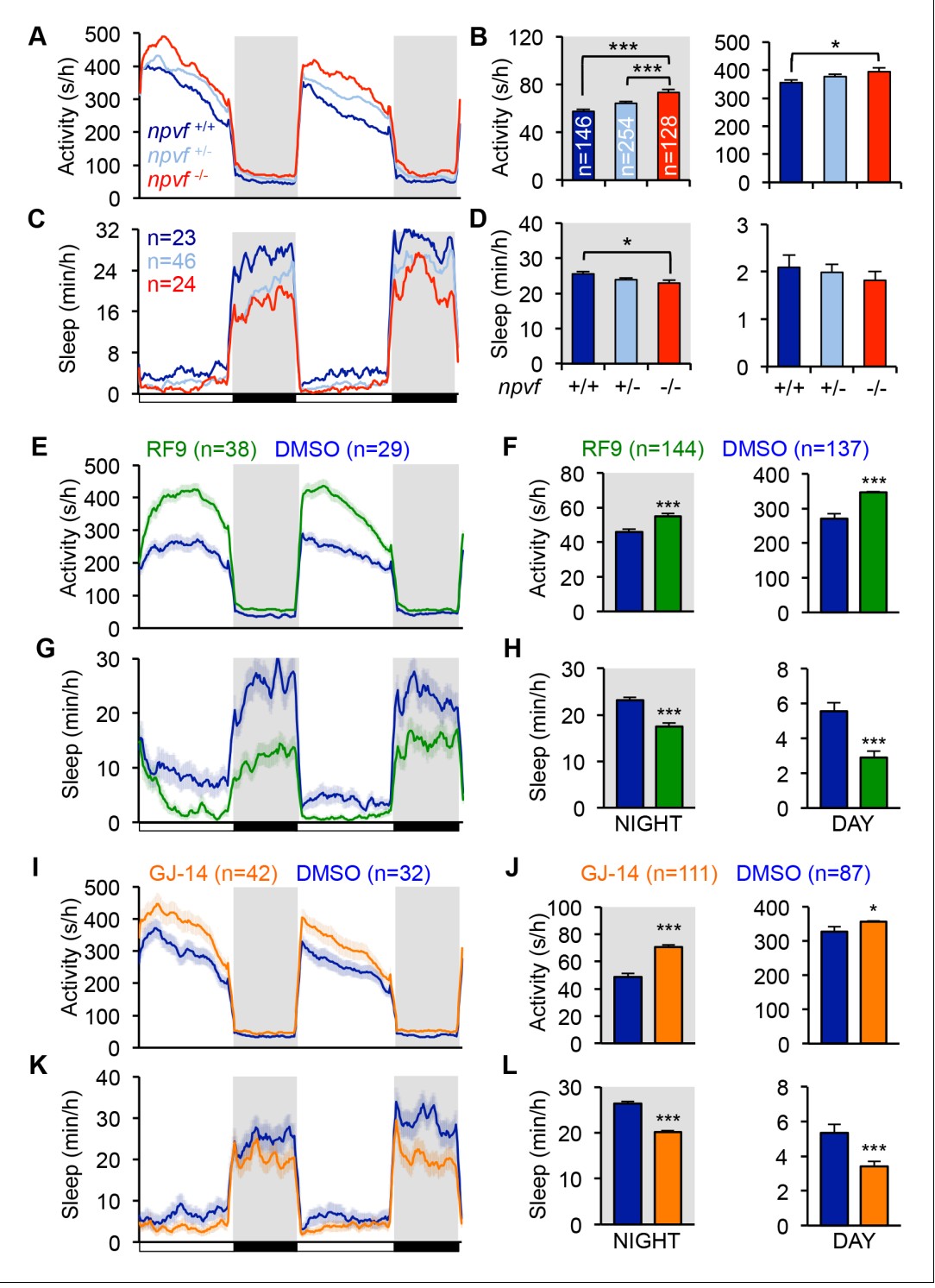

**Figure 3.** Genetic and pharmacological inhibition of NPVF signaling increases locomotor activity and decreases sleep. (A–D) *npvf* -/- animals are more active during the day and night (A,B), and sleep less at night (C,D) compared to their *npvf* +/+ siblings. *npvf* +/- animals show an intermediate sleep phenotype that is not significantly different than *npvf* -/- siblings. (E–L) Pharmacological inhibition of NPVF signaling by the NPFFR1/2 antagonists RF9 and GJ-14 increased locomotor activity (E,F,I,J) and decreased sleep (G,H,K,L) during the day and night compared to DMSO control-treated siblings. Mean ± SEM from one representative experiment (A,C,E,G,L, K), three pooled experiments (F,H,J,L), or six pooled experiments (B,D) are shown. White and black bars under behavioral traces indicate day (14 h) and night (10 h), respectively. n = number of animals. *p<0.05; ***p<0.005 by

*Figure 3 continued on next page*

*Figure 3 continued*

One-way ANOVA with Holm-Sidak test (B,D) and Student's *t*-test (F,H,J,L). See also *Figure 3—figure supplements 1* and *2*.

DOI: https://doi.org/10.7554/eLife.25727.009

The following figure supplements are available for figure 3:

**Figure supplement 1.** Genetic and pharmacological inhibition of NPVF signaling alters sleep/wake architecture.
DOI: https://doi.org/10.7554/eLife.25727.010

**Figure supplement 2.** Pharmacological inhibition of NPVF signaling suppresses RFRP-overexpression induced sleep.
DOI: https://doi.org/10.7554/eLife.25727.011

## NPVF signaling affects arousal threshold

Sleep is distinguished from quiet wakefulness by an increased arousal threshold, which we assayed by delivering mechano-acoustic stimuli of variable intensities to zebrafish larvae while monitoring their behavior (*Gandhi et al., 2015*). We monitored the fraction of animals that responded to the stimulus at several intensities and used this data to construct dose-response curves. Based on these curves, we calculated the tapping intensity at which the half-maximal response occurred (effective tap power 50, $ETP_{50}$). We observed that the $ETP_{50}$ for NPVF-overexpressing larvae was 227% higher than that of their WT siblings (*Figure 4A*) ($p<0.0001$ by extra sum-of-squares F test). Conversely the $ETP_{50}$ for RF9- or GJ-14-treated animals was 46% or 66% lower than that of their DMSO-treated siblings, respectively (*Figure 4B and C*) (both: $p<0.0001$ by extra sum-of-squares F test). These results demonstrate that NPVF overexpression increases arousal threshold, whereas inhibition of NPVF signaling decreases arousal threshold.

## Identification of a genomic element that drives gene expression in NPVF neurons

In mammalian brains, *npvf* is highly enriched in the dorsomedial hypothalamus (*Hinuma et al., 2000*; *Liu et al., 2001*; *Ubuka et al., 2009*). Similarly, using ISH we found that *npvf* is exclusively expressed in the zebrafish dorsomedial hypothalamus at 5 days post-fertilization (dpf) (*Figure 5A–C*), similar to previous observations (*Madelaine et al., 2017*; *Yelin-Bekerman et al., 2015*). However, zebrafish larvae only have ~15 NPVF neurons per brain hemisphere, over an order of magnitude fewer than

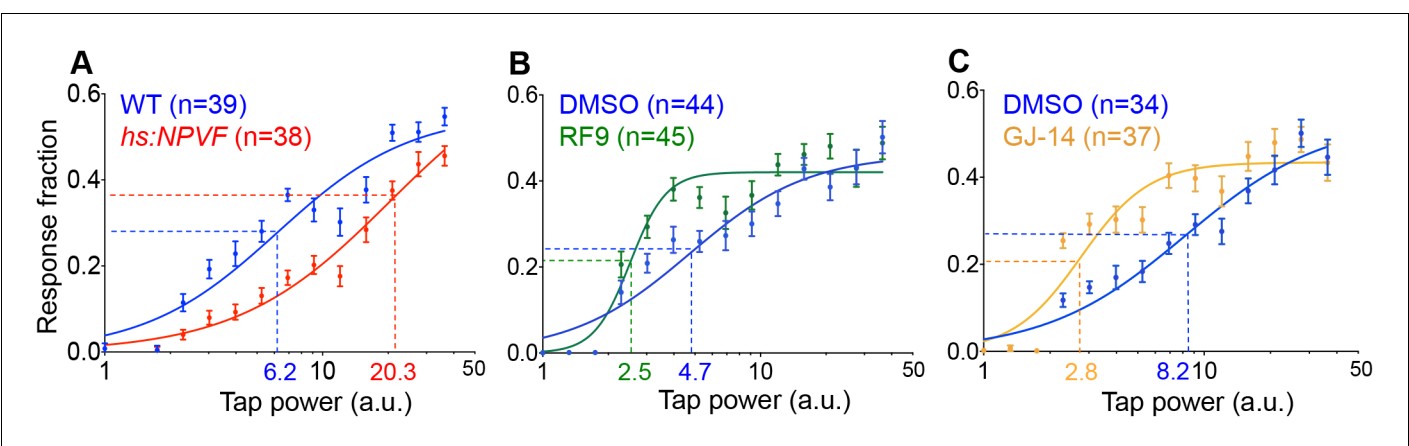

**Figure 4.** NPVF signaling affects arousal threshold. Representative stimulus response curves generated using a mechano-acoustic stimulus in *Tg(hs: NPVF)* animals and their WT siblings following heat shock (A), in RF9 and DMSO vehicle treated WT larvae (B), and in GJ-14 and DMSO vehicle treated WT larvae (C). Each data point indicates mean ± SEM. Dashed lines mark the $ETP_{50}$ value for each genotype or drug treatment. (A) *Tg(hs:NPVF)* animals had an $ETP_{50}$ value of 20.3 vs. 6.2 for WT siblings (227% increase, $F_{(1,834)}=16.05$, $p<0.0001$ by extra sum-of-squares F test). (B) RF9-treated animals had an $ETP_{50}$ of 2.5 vs. 4.7 for DMSO-treated siblings (46% decrease, $F_{(1,834)}=24.19$, $p<0.0001$ by extra sum-of-squares F test). (C) GJ14-treated animals had an $ETP_{50}$ of 2.8 vs. 8.2 for DMSO-treated siblings (65% decrease, $F_{(1,834)}=49.47$, $p<0.0001$ by extra sum-of-squares F test). n = number of animals.
DOI: https://doi.org/10.7554/eLife.25727.012

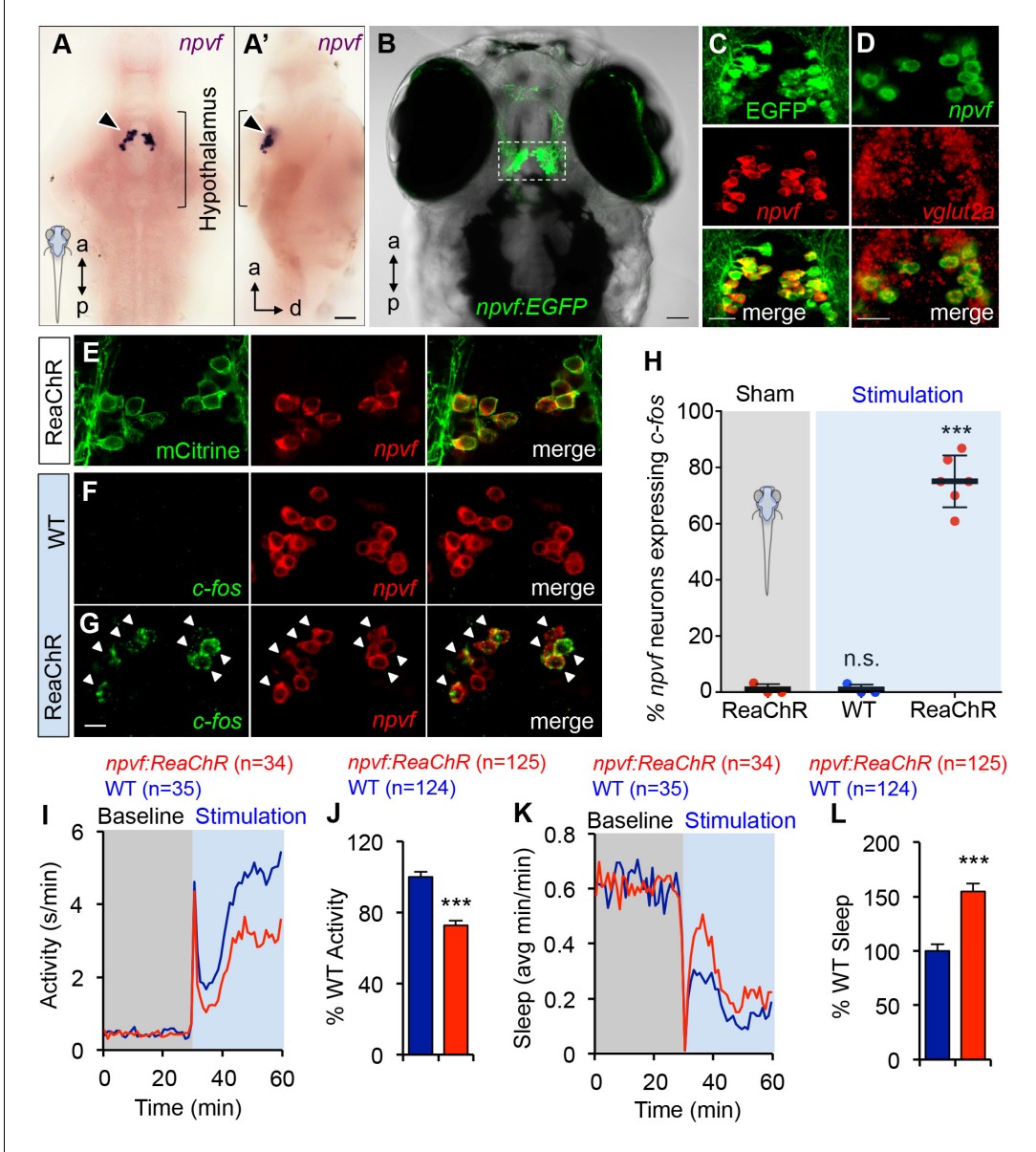

**Figure 5.** Optogenetic stimulation of NPVF neurons promotes sleep. (**A**) ISH of NPVF neurons (arrowhead) in a 5-dpf brain. Ventral (**A**) and side (**A'**) views are shown. a, anterior; p, posterior; d, dorsal. (**B**) Live image of a 5-dpf *Tg(npvf:EGFP)* zebrafish with brightfield and EGFP overlay. White box demarcates region shown in subsequent images. (**C**) EGFP and *npvf* coexpression shown using EGFP immunohistochemistry (IHC) and *npvf* FISH in a 43.4 μm thick image stack. (**D**) *npvf* and *vglut2a* coexpression shown using FISH in a 2.7 μm thick image stack. (**E**) mCitrine and *npvf* coexpression in a *Tg(npvf:ReaChR-mCitrine)* zebrafish shown using EGFP IHC and *npvf* FISH in a 2.7 μm thick image stack. (**F–H**) *Tg(npvf:ReaChR-mCitrine)* and WT siblings were exposed to the same blue light stimulus used in (**I–L**). (**G,H**) 75.1 ± 4.1% of NPVF neurons expressed *c-fos* in *Tg(npvf:ReaChR-mCitrine)* animals (***p<0.0001; One-way ANOVA with Dunnett's test to sham), in contrast to similarly stimulated WT siblings (~1%, n.s., not significant) (**F,H**), or to sham-treated transgenic animals (~1%). (**I–L**) Optogenetic stimulation of NPVF neurons decreased locomotor activity and increased sleep in transgenic animals compared to WT siblings. Data are from one representative experiment (**I,K**), or three pooled experiments (**J,L**). Activity and sleep of transgenic animals are normalized to WT and represented as mean ± SEM (**J,L**). n = number of animals. ***p<0.0001 by Student's t-test. Scale: 50 μm (**A,B**), 20 μm (**C,D**), 10 μm (**G**). See also *Table 1* and *Figure 5—figure supplements 1,2*.

DOI: https://doi.org/10.7554/eLife.25727.013

The following figure supplements are available for figure 5:

**Figure supplement 1.** NPVF neuronal projections.
DOI: https://doi.org/10.7554/eLife.25727.014

**Figure supplement 2.** Chemogenetic stimulation of NPVF neurons promotes sleep.
DOI: https://doi.org/10.7554/eLife.25727.015

the number of NPVF neurons in mice (*Poling et al., 2012*). Using fluorescent *in situ* hybridization (FISH), we observed that *npvf*-expressing neurons also express the glutamatergic markers *vglut2a* (*Figure 5D*, 95 ± 3%, n = 4) and *vglut2b* (83 ± 3%, n = 6), and do not express the GABAergic marker *gad67* (data not shown, n = 3), suggesting that this neuronal population is excitatory in nature. We further identified a 4 kb genomic region directly 5′ to the *npvf* open reading frame that is sufficient to drive expression of enhanced green fluorescent protein (EGFP) specifically in *npvf*-expressing neurons, with 98 ± 1% (n = 12) of EGFP positive cells expressing *npvf* mRNA, and 97 ± 1% (n = 12) of *npvf*-positive cells expressing EGFP (*Figure 5B and C*, *Table 1*). This neuronal population sends dense projections within the hypothalamus and to the raphe nuclei of the hindbrain (*Madelaine et al., 2017*), and sparse projections to the forebrain, tectum, and spinal cord (*Figure 5—figure supplement 1*), suggesting that it may regulate neurons in these regions (*Madelaine et al., 2017*).

## Optogenetic stimulation of NPVF neurons promotes sleep

To test the hypothesis that NPVF neurons are sufficient to promote sleep, we generated transgenic animals in which these neurons express the channelrhodopsin variant ReaChR (*Lin et al., 2013*) fused to mCitrine (*Figure 5E* and *Table 1*). To determine whether this transgene can stimulate NPVF neurons, we used a non-invasive, large-scale assay (*Singh et al., 2015*) in which freely-moving 5 dpf animals were exposed to blue light (470 nm at ~400 μW) for 30 min, and then performed double FISH using probes specific for *npvf* and *c-fos*. We observed that 75% of *npvf*-expressing neurons were *c-fos* positive in *Tg(npvf:ReaChR-mCitrine)* animals compared to ~1% in identically treated WT siblings or in transgenic animals not exposed to blue light (sham; *Figure 5F–H*).

We next tested the behavioral effect of stimulating NPVF neurons using the same behavioral assay. *Tg(npvf:ReaChR-mCitrine)* animals and their WT siblings exhibited similar baseline locomotor activity levels. Upon blue light exposure, both transgenic and WT siblings responded with an approximately 30 s burst of activity that we exclude from analysis, followed by a return to near baseline activity levels, and a gradual increase in activity that plateaus after ~15 min (*Figure 5I*). During blue light stimulation, locomotor activity was 27% lower (*Figure 5I and J*) and sleep was 55% higher (*Figure 5K and L*) for *Tg(npvf:ReaChR-mCitrine)* animals compared to their WT siblings (activity and sleep: p<0.0001, Student's *t*-test). Thus, stimulation of NPVF neurons suppresses locomotor activity and promotes sleep compared to non-transgenic controls, consistent with the NPVF overexpression phenotype.

**Table 1.** Specificity of transgenic lines used in this study.
Specificity of transgenic lines was confirmed by FISH and/or IHC on dissected whole mount 5-dpf zebrafish brains. Each brain hemisphere contains ~15 NPVF neurons. All neurons were counted in each hemisphere and percentage of co-expression is displayed as mean ± SEM.

| Transgenic line | Sample size (brain hemispheres) | Fluorescent-neurons co-expressing *npvf* | *npvf* neurons co-expressing fluorescent-tagged transgene |
| --- | --- | --- | --- |
| *Tg(npvf:EGFP)* | 12 | 98.2 ± 1.0% | 97.2 ± 1.1% |
| *Tg(npvf:ReaChR-mCitrine)* | 12 | 97.1 ± 1.4% | 94.1 ± 2.3% |
| *Tg(npvf:mTagYFP-T2A-eNTR)* | 6 | 96.9 ± 2.3% | 97.7 ± 1. 6% |
| Transgenic Line | Sample size (brain hemispheres) | Red fluorescent neurons co-expressing EGFP | EGFP-expressing NPVF neurons co-expressing red fluorescence |
| *Tg(npvf:KalTA4); Tg(UAS:nfsb-mCherry); Tg(npvf:EGFP)* | 16 | 100 ± 0.0% | 100 ± 0.0% |
| *Tg(npvf:KalTA4); Tg(UAS:TRPV1-TagRFP-T); Tg(npvf:EGFP)* | 4 | 88.8 ± 4.5% | 92.3 ± 4.4% |

DOI: https://doi.org/10.7554/eLife.25727.016

## Chemogenetic stimulation of NPVF neurons promotes sleep

As an alternative method to stimulate NPVF neurons that does not require a light stimulus, we used an approach in which neurons expressing the rat TRPV1 channel are activated following addition of its small molecule agonist capsaicin (Csn) to the water (*Chen et al., 2016c*). We generated *Tg(npvf: KalTA4)* transgenic animals, in which NPVF neurons express an optimized form of the transcriptional activator Gal4 (*Distel et al., 2009*), and *Tg(UAS:TRPV1-TagRFP-T)* transgenic animals, in which TRPV1 expression is induced in KalTA4-expressing cells (*Figure 5—figure supplement 1A*, *Table 1*) (*Chen et al., 2016c*). Following treatment with 2 µM Csn, *Tg(npvf:KalTA4); Tg(UAS:TRPV1-TagRFP-T); Tg(npvf:EGFP)* animals displayed higher levels of *c-fos* expression in NPVF neurons compared to identically treated *Tg(npvf:KalTA4); Tg(npvf:EGFP)* siblings (*Figure 5—figure supplement 1F G*), consistent with previous results demonstrating that Csn stimulates TRPV1-expressing neurons *in vivo* (*Chen et al., 2016c*). Because higher concentrations of Csn (≥10 µM) can result in apoptosis (*Chen et al., 2016c*), we compared the number of NPVF neurons in *Tg(npvf:KalTA4); Tg(UAS: TRPV1-TagRFP-T)* animals to their *Tg(npvf:KalTA4)* siblings after treatment with 2 µM Csn. We observed the same number of NPVF neurons for both genotypes, indicating that 2 µM Csn did not result in the loss of TRPV1-expressing NPVF neurons (*Figure 5—figure supplement 1L M*).

We next tested the behavioral effect of stimulating TRPV1-expressing NPVF neurons. *Tg(npvf: KalTA4); Tg(UAS:TRPV1-TagRFP-T)* and *Tg(npvf:KalTA4)* siblings treated with DMSO vehicle control displayed no differences in locomotor activity or sleep (*Figure 5—figure supplement 1B–E*). In contrast, upon treatment with 2 µM Csn, *Tg(npvf:KalTA4); Tg(UAS:TRPV1-TagRFP-T)* animals exhibited 20% more sleep and 13% less locomotor activity at night compared to *Tg(npvf:KalTA4)* siblings (both p<0.001; Student's *t*-test) (*Figure 5—figure supplement 1H–K*). While the behavioral phenotype induced using optogenetic stimulation of NPVF neurons was larger than that using TRPV1, optogenetic stimulation induced more robust *c-fos* expression than TRPV1-dependent stimulation (*Figure 5G* and *Figure 5—figure supplement 2F*), suggesting that more robust stimulation of NPVF neurons was achieved using optogenetics. The different level of stimulation might result from different transgene expression levels, acute stimulation with ReaChR versus prolonged stimulation with TRPV1, or different effects of ReaChR and TRPV1 on neuronal physiology. Taken together, these data show that chemogenetic stimulation of NPVF neurons promotes sleep, consistent with both the NPVF optogenetic and overexpression data.

## Chemogenetic ablation of NPVF neurons promotes wakefulness

To determine whether NPVF neurons are required for normal sleep levels, we generated *Tg(npvf: mTagYFP-T2A-eNTR)* transgenic animals, in which NPVF neurons express membrane targeted TagYFP and enhanced nitroreductase (eNTR) (*Mathias et al., 2014*; *Tabor et al., 2014*), which converts the inert pro-drug metronidazole (MTZ) into a DNA cross-linking agent that induces cell-autonomous death (*Figure 6A* and *Table 1*). MTZ treatment of *Tg(npvf:mTagYFP-T2A-eNTR)* animals resulted in nearly complete loss of NPVF neurons (*Figure 6B and C*). During both the day and night, MTZ-treated transgenic animals were more active (+29% and +44%, respectively, both: p<0.005, Student's *t*-test) and slept less (−49% and −10%, respectively, both: p<0.005, Student's *t*-test) than their MTZ-treated non-transgenic siblings (*Figure 6D–6G*). NPVF neuron-ablated animals exhibited fewer sleep bouts (day: −43%; night: −9%, both p<0.0001, Student's *t*-test), increased wake activity (day: +22%; night: +20%, both: p<0.0001, Student's *t*-test), and increased sleep latency at night (+25%, p<0.005, Student's *t*-test) compared to non-transgenic sibling controls (*Figure 6—figure supplement 1A–D*). These results indicate that NPVF neurons are necessary for both the initiation and maintenance of normal sleep levels during the day and night, consistent with the phenotype resulting from pharmacological inhibition of NPVF signaling (*Figure 3* and *Figure 3—figure supplement 1F–I*).

## NPVF neuronal stimulation results in neuronal activity levels similar to those normally observed at night

As an independent demonstration that NPVF neurons promote sleep, we developed an intravital imaging assay of neuronal activity in the brain during normal sleep/wake states and in response to stimulation of NPVF neurons. Two-photon selective plane illumination microscopy (2P-SPIM) (*Truong et al., 2011*) was used to image neuronal activity in *Tg(elavl3:H2B-GCaMP6s)* animals, which

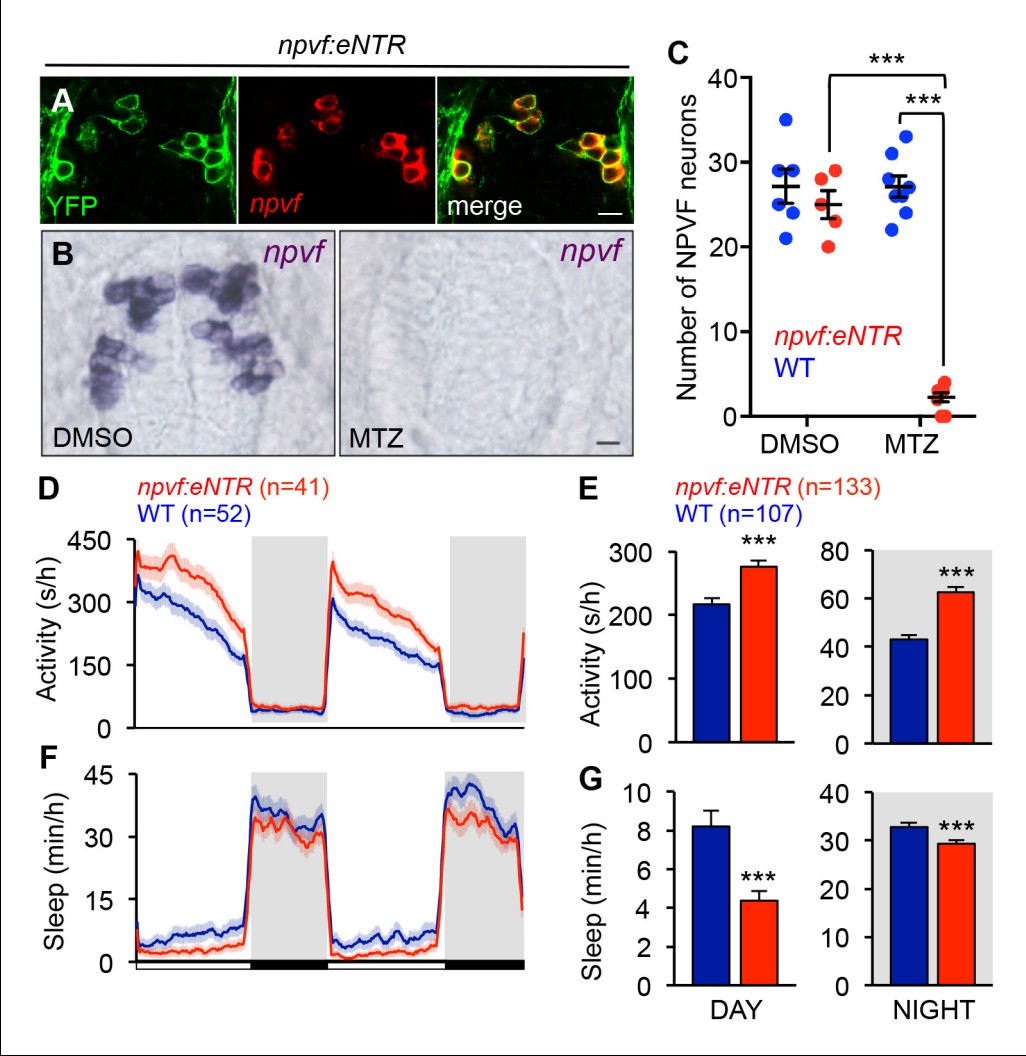

**Figure 6.** Chemogenetic ablation of NPVF neurons promotes wakefulness. (**A**) mTagYFP and *npvf* coexpression in a 5-dpf *Tg(npvf:mTagYFP-T2A-eNTR)* zebrafish shown using YFP IHC and *npvf* FISH. (**B–G**) *Tg(npvf:mTagYFP-T2A-eNTR)* and WT siblings were treated with 5 mM MTZ for 60 h. Behavior was monitored from 6 to 8 dpf. (**B**) *npvf* ISH in 7-dpf *Tg(npvf:mTagYFP-T2A-eNTR)* zebrafish treated with DMSO or MTZ. Scale: 10 μm. NPVF neuronal loss is quantified as mean ± SEM in (**C**). (**D–G**) Transgenic animals treated with MTZ were more active (**D,E**) and slept less (**F,G**) than their identically treated WT siblings. Mean ± SEM for one representative experiment (**D,F**), or three pooled experiments (**E,G**) are shown. White and black bars under behavioral traces indicate day (14 h) and night (10 h), respectively. n = number of animals. ***p<0.005 by Two-way ANOVA with Holm-Sidak test (**C**) or Student's *t*-test (**E,G**). See also *Table 1* and *Figure 6—figure supplement 1*.

DOI: https://doi.org/10.7554/eLife.25727.017

The following figure supplement is available for figure 6:

**Figure supplement 1.** Chemogenetic ablation of NPVF neurons alters sleep/wake architecture.

DOI: https://doi.org/10.7554/eLife.25727.018

express the genetically encoded calcium indicator GCaMP6s in most neurons (*Vladimirov et al., 2014*). 2P-SPIM combines the high-speed and low photodamage of light sheet microscopy with the high penetration depth and invisibility of 2P excitation light, which avoids visible light-induced perturbation of sleep. We entrained the animals in 13:11 h light:dark conditions until 5 dpf, and then imaged GCaMP6s fluorescence in constant dark for over 36 h (at 1 Hz for 2 min every 15 min); GCaMP6s fluorescence was measured in a region of the tectum due to its accessibility and suitability for long-term imaging (*Figure 7A*, rectangle). To measure arousal, during each 2 min imaging trial

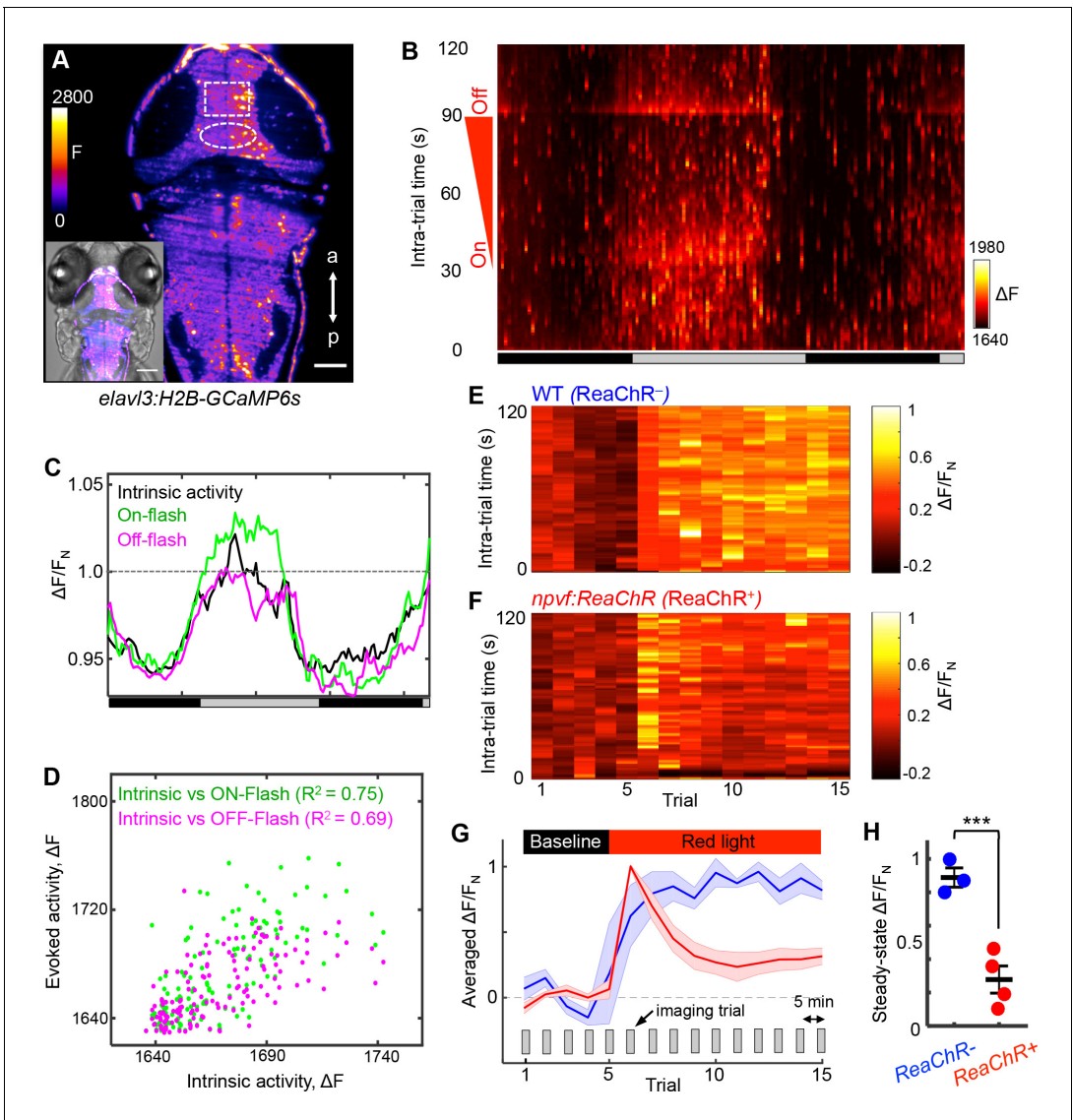

**Figure 7.** NPVF neuronal activation induces tectal neuronal activity levels similar to normal sleep. (**A**) Representative 2P-SPIM image of a 6 dpf *Tg (elavl3:H2B-GCaMP6s)* animal, ~70 μm from the dorsal surface. The white box indicates the region of the optic tectum analyzed in subsequent panels. The white oval indicates the location of NPVF neurons. The inset shows a GCaMP6s image superimposed on a brightfield image. Scale: 50 μm. (**B**) Neural activity, recorded as GCaMP6s fluorescence, of a representative animal during >36 h of imaging in DD. Imaging sessions were performed for 2 min at 1 Hz every 15 min. No light perturbation was applied during the first 31 s of each trial to record intrinsic brain activity. To assess arousal, a red light was turned on at t = 32 s, gradually increased in intensity until t = 93 s, then turned off. We observed increased neural activity at light onset and offset, with habituation in between. Signal is calculated as $\Delta F = F - F_0$ where $F_0$ = lowest signal measured during the recording period. (**C**) Normalized neural activity, plotted as function of time, smoothed over a 5-trial sliding window, and averaged over 5–25, 32–37, and 93–98 s time windows, representing intrinsic, On-flash evoked response and Off-flash evoked response, respectively. Signal is normalized by $F_N$ = average intrinsic activity during subjective day. (**D**) Correlation of evoked activity to intrinsic activity. (**E,F**) During subjective day, animals were imaged for 15 trials, each consisting of 2 min of imaging at 1 Hz every 5 min. Conditions were identical to those used in (**B**). Representative neural activity recorded for individual ReaChR- and ReaChR+ animals are shown. Optogenetic stimulation began after trial 5. Signal for each animal is calculated with baseline $F_0$ = averaged signal over the first five trials, then normalized by $F_N$ = peak of signal after red light is turned on, for the same animal. (**G**) Mean ± SEM of the normalized neural activity for four *Tg(elavl3:H2B-GCaMP6s); Tg(npvf:ReaChR-mCitrine)* larvae (red) and three *Tg(elavl3:H2B-GCaMP6s)* siblings (blue). (**H**) Mean ± SEM steady-state normalized neural activity after optogenetic activation, averaged over trials 10–15, shows 68% lower activity for ReaChR + animals than their ReaChR- siblings (p<0.001, two-sample *t*-test). See also *Figure 7—figure supplements 1* and *2*.

DOI: https://doi.org/10.7554/eLife.25727.019

The following figure supplements are available for figure 7:

**Figure supplement 1.** Effect of optogenetic stimulation of NPVF neurons on GCaMP6s fluorescence in the tectum of individual animals.

DOI: https://doi.org/10.7554/eLife.25727.020

*Figure 7 continued on next page*

*Figure 7 continued*

**Figure supplement 2.** Chemogenetic stimulation of NPVF neurons decreases neuronal activity at night.
DOI: https://doi.org/10.7554/eLife.25727.021

we applied an ambient red light of increasing intensity (on: t = 32 s; off: t = 93 s) (*Figure 7B*). Thus, during each trial we imaged intrinsic activity (t = 0–31 s), as well as responses evoked by lights on (t = 32 s) and lights off (t = 93 s). Both intrinsic and evoked activity followed the subjective day/night cycle, with higher levels during subjective day (*Figure 7B and C*), and correlated magnitudes of intrinsic and evoked responses (*Figure 7D*). This pattern of intrinsic activity is consistent with sleep/wake oscillations. The decreased responsiveness to a stimulus during subjective night is consistent with the increased arousal threshold that occurs during sleep, although it could also result from processes regulated by the circadian clock (*Emran et al., 2010*). Thus, this assay allows long-term monitoring of neuronal activity in the brain during what appear to be normal day/night patterns of neuronal activity and arousal levels.

To test whether sleep induced by activating NPVF neurons produces similar effects on neuronal activity, we stimulated these neurons using ReaChR with focused red light while imaging GCaMP6s fluorescence in a region of the tectum during subjective day. As expected for a visual response to red light, fluorescence in the tectum of ReaChR-negative animals increased during the period of light exposure (*Figure 7E and G* and *Figure 7—figure supplement 1A*). In contrast, after a transient response to the focused red light, *Tg(npvf:ReaChR-mCitrine)* animals displayed a suppression of neuronal activity to levels approaching those normally observed at night (*Figure 7F–H* and *Figure 7—figure supplement 1B*). Suppression of tectal neuronal activity in the absence of visual stimuli during subjective night, and in response to NPVF neuron stimulation, is surprising because vertebrate sensory systems are thought to be activated similarly during sleep and wake states, with the thalamus limiting responses to these stimuli during sleep (*Coulon et al., 2012*). Our data suggest that sleep-promoting mechanisms may act upstream of the thalamus to inhibit brain regions that receive direct input from sensory systems.

## Chemogenetic stimulation of NPVF neurons suppresses neuronal activity throughout the brain

As an alternative approach to stimulate NPVF neurons while monitoring neuronal activity in the brain using 2P-SPIM, we compared GCaMP6s fluorescence in *Tg(elavl3:H2B-GCaMP6s)* animals that expressed TRPV1 in NPVF neurons (*Tg(npvf:KalTA4); Tg(UAS:TRPV1-TagRFP-T))* to siblings that did not (*Tg(UAS:TRPV1-TagRFP-T))* during exposure to 2 µM Csn in 13:11 h light:dark conditions. To analyze neuronal responses in a more comprehensive manner, we used principal component and independent component analysis (PCA/ICA) to automatically segment 500–800 spontaneously active neurons throughout a single optical section of the brain (*Figure 7—figure supplement 2A and B*). Using this approach, we observed lower average levels of neuronal activity at night compared to the day for *Tg(UAS:TRPV1-TagRFP-T)* control animals (7% decrease, p=0.023, one-sample *t*-test) (*Figure 7—figure supplement 2C and E–G*), similar to the results described above in the optic tectum (*Figure 7B and C*). Stimulation of NPVF neurons in *Tg(npvf:KalTA4); Tg(UAS:TRPV1-TagRFP-T)* animals had no significant effect on neuronal activity during the day compared to *Tg(UAS:TRPV1-TagRFP-T)* sibling controls (p=0.68, two-sample *t*-test) (*Figure 7—figure supplement 2F*). However, stimulation of NPVF neurons in *Tg(npvf:KalTA4); Tg(UAS:TRPV1-TagRFP-T)* animals dramatically suppressed neuronal activity at night compared to the day (25% decrease, p=0.023, one-sample *t*-test) (*Figure 7—figure supplement 2D–G*), and compared to *Tg(UAS:TRPV1-TagRFP-T)* siblings at night (~3 fold difference, p=0.028, two-sample *t*-test) (*Figure 7—figure supplement 2G*). This result indicates that stimulation of NPVF neurons using TRPV1 suppresses neuronal activity in the brain at night, consistent with the increased sleep at night observed upon stimulation of NPVF neurons using TRPV1 (*Figure 5—figure supplement 2*), and with our results using optogenetic stimulation of NPVF neurons (*Figure 5*). We conclude that stimulation of NPVF neurons is sufficient to both increase sleep and decrease neuronal activity in the brain.

## Discussion

Identifying genes that regulate vertebrate sleep has been hampered, in part, by the difficulty of performing genetic screens for behavioral phenotypes in rodent animal models. In contrast, genetic and genomic screens using invertebrate models, such as flies and nematodes, have identified many genes that regulate sleep (*Cirelli et al., 2005*; *Iannacone et al., 2017*; *Koh et al., 2008*; *Nath et al., 2016*; *Shang et al., 2013*; *Shi et al., 2014*). For example, in *Drosophila*, loss of function of the RFamide neuropeptide sNPF reduces sleep, whereas stimulation of sNPF-expressing neurons promotes sleep (*Shang et al., 2013*). Similarly, in *C. elegans*, overexpression of several members of the FLP family of RFamide neuropeptides promotes sleep (*Nath et al., 2016*; *Nelson et al., 2014*; *Trojanowski et al., 2015*), while loss of function analysis revealed a requirement for select FLP neuropeptides in regulating sleep duration (*Chen et al., 2016b*; *Iannacone et al., 2017*; *Nath et al., 2016*; *Nelson et al., 2014*; *Turek et al., 2016*) or homeostasis (*Nagy et al., 2014*). Based on these data, we reasoned that testing candidate vertebrate orthologs of invertebrate sleep regulators may be a fruitful strategy to identify neuropeptides that regulate vertebrate sleep. Here we used this approach to identify a vertebrate RFamide neuropeptide sleep-promoting system.

We found that signaling by neuropeptides derived from the vertebrate NPVF preproprotein, as well as NPVF neurons, are both necessary and sufficient to promote sleep in zebrafish larvae. Similar to *C. elegans* FLP-13 (*Nath et al., 2016*), we found that robust NPVF overexpression-induced sleep requires the synergistic interaction of two or more RFRP peptides derived from the NPVF preproprotein. While invertebrate RFamide neuropeptides share some amino acid similarity with vertebrate RFRP peptides, it was unclear to what extent invertebrate neuropeptidergic sleep mechanisms are conserved in vertebrates. Indeed, the conserved role of RFamide neuropeptides in promoting sleep is surprising considering that previously described neuropeptidergic regulators of vertebrate sleep, such as Hcrt (*Chemelli et al., 1999*; *Lin et al., 1999*) and MCH (*Willie et al., 2008*), lack clear invertebrate orthologs. We recently showed that another vertebrate RFamide peptide, QRFP, promotes sleep in zebrafish (*Chen et al., 2016a*), but its effects are much weaker than those of NPVF in zebrafish, and of FLP-13 in *C. elegans* (*Nath et al., 2016*; *Nelson et al., 2014*). Consistent with these observations, the amino acid sequence of the C-terminus of the zebrafish QRFP peptide (GGFGFRF) is less similar than those of the zebrafish RFRP peptides (LP[L/Q]RF) to the *C. elegans* FLP-13 peptides (P[L/F]IRF). Thus, to our knowledge, this study provides the first example of a neuropeptide family that has a strong role in regulating sleep in both vertebrates and invertebrates, suggesting that it participates in an ancient and central aspect of sleep control.

Consistent with an important role for NPVF in regulating sleep is its prominent expression in the hypothalamus, an ancient brain structure that is anatomically and molecularly conserved in mammals and zebrafish (*Biran et al., 2015*; *Liu et al., 2015*; *Shimogori et al., 2010*). Classical lesion studies found that the hypothalamus plays a critical role in promoting both sleep and wakefulness (*Saper et al., 2010*; *Weber and Dan, 2016*). More recently, specific hypothalamic wake- and sleep-promoting neuronal populations have been identified (*Adamantidis et al., 2007*; *Chung et al., 2017*; *Konadhode et al., 2013*; *Singh et al., 2015*; *Tsunematsu et al., 2014*; *Verret et al., 2003*; *Willie et al., 2008*). Some hypothalamic neurons are also proposed to regulate a switch between sleep and wake states that are driven by neurons in the hypothalamus and brainstem, and affect neurons in the forebrain (*Saper et al., 2010*). We found that stimulation or ablation of zebrafish NPVF neurons resulted in increased sleep or wake, respectively, demonstrating that these neurons participate in hypothalamic regulation of sleep. We also observed that zebrafish NPVF neurons send dense projections to hypothalamic and brainstem regions, as well as sparse projections to the forebrain. Therefore, the anatomical location and projection pattern of NPVF neurons, combined with our behavioral results, suggests that these neurons may act to integrate cues relevant to sleep and affect hypothalamic, hindbrain and forebrain neurons that regulate sleep/wake states.

Studies in invertebrate organisms suggest that RFamide-expressing neurons may have arisen from cells with dual sensory-neurosecretory properties that were the starting point for the evolution of neurosecretory brain centers in Bilateria (*Plickert and Schneider, 2004*; *Tessmar-Raible et al., 2007*). In *C. elegans*, the neurosecretory ALA neuron produces several sleep-promoting RFamide neuropeptides including FLP-13 (*Nath et al., 2016*; *Nelson et al., 2014*), and is required for sleep that is induced by cellular stressors, including noxious temperature, hypertonicity and tissue damage (*Hill et al., 2014*; *Nath et al., 2016*). It remains unclear whether cellular stress induces sleep in

vertebrates, but the increased sleep in observed in mammals during illness and following cellular injury might be regulated by mechanisms analogous to those that promote cellular stress-induced sleep in *C. elegans* (*Davis and Raizen, 2017*). We speculate that NPVF neurons may have a sensory function to detect physiological changes such as cellular stress, integrate this information, and promote a behavioral state change from wake to sleep. Indeed, NPVF neurons project widely in the hypothalamus and hindbrain (*Figure 5—figure supplement 1*) suggesting they receive input from and influence large regions of the brain. These include the optic tectum (*Figure 5—figure supplement 1* and *Figure 7*), which processes light cues from the retina that may influence sleep, and brain ventricular cells (*Figure 2—figure supplement 2*), which secrete factors into the cerebrospinal fluid that can promote sleep (*Krueger et al., 2007*). This possibility can be addressed by identifying both upstream and downstream mechanisms involved in NPVF-regulated sleep in the vertebrate brain, and can also be informed by examining potentially homologous neurons in simpler species.

Most genes implicated in regulating sleep also affect other physiological and behavioral processes. For example, Hcrt not only promotes arousal, but also regulates feeding, energy homeostasis, reward seeking and addiction (*Li et al., 2016*). Previous studies of RFRP peptides in birds and rodents have demonstrated roles in regulating reproductive behaviors and nociception (*Kim, 2016*; *Kriegsfeld et al., 2015*; *Tsutsui et al., 2010*; *Tsutsui et al., 2012*; *Tsutsui et al., 2015*; *Ubuka et al., 2014*), and a study in zebrafish implicated NPVF neurons in nociception (*Madelaine et al., 2017*), but a role in sleep has not been reported. The rodent and avian studies mostly used infusion of single, *in vitro* synthesized RFRP peptides that, in addition to potentially lacking post-translational modifications that may be necessary for normal function, do not take into consideration the combinatorial action of RFRP peptides to promote sleep. We found that genetic overexpression of at least two distinct RFRP peptides is required to induce robust sleep in zebrafish. Thus, similar to nematode FLP peptides (*Nath et al., 2016*), distinct RFRP peptides may work together to control vertebrate sleep by inhibiting select behaviors associated with arousal, such as reproduction (*Tsutsui et al., 2015*; *Ubuka et al., 2014*) and feeding (*Ubuka et al., 2016*). Consistent with our findings, knockdown of *npvf* expression in Japanese quail (*Ubuka et al., 2014*) and the white-crowned sparrow (*Ubuka et al., 2014*) increases locomotor activity. However, genetic gain- or loss-of-function analyses have not been reported for *npvf* or its receptors in any other species, and a function for NPVF neurons in regulating sleep had not previously been described.

Several aspects of the NPVF system make it a particularly interesting sleep regulator. First, zebrafish NPVF neurons are glutamatergic, and thus excitatory, in contrast to most previously described sleep-promoting neurons, which are GABAergic, and thus inhibitory (*Richter et al., 2014*; *Saper et al., 2005*). For example, the *C. elegans* sleep promoting neuron ALA, which produces FLP-13, was recently shown to be GABAergic (*Gendrel et al., 2016*), suggesting a difference in how neurons that produce FLP-13 in *C. elegans* and NPVF in zebrafish interact with the sleep neuronal circuitry. However, the excitatory nature of NPVF neurons in zebrafish is consistent with observations in mammals, as RNA sequencing found that *npvf*-expressing neurons are glutamatergic in mice (*Chen et al., 2017*), and NPVF neurons do not co-express the vesicular GABA transporter in mice (*Rizwan et al., 2014*). Thus, NPVF neurons may regulate sleep via a neural circuit mechanism that is distinct from previously described sleep-promoting neurons.

Second, the zebrafish and human NPVF orthologs contain three RFRP peptides (*Ubuka and Tsutsui, 2014*) in contrast to rodent and avian species, which lack RFRP2 (*Hinuma et al., 2000*; *Liu et al., 2001*). The zebrafish may thus be a more appropriate model than rodents and birds for some aspects of human NPVF function. The conservation of RFRP2 between zebrafish and humans is particularly important, because while we observed that overexpression of at least two RFRP peptides is required to induce robust sleep, overexpression of RFRP2 alone was sufficient to significantly increase sleep. These observations suggest that RFRP2 has stronger sleep-promoting effects than RFRP1 or RFRP3. Consistent with these results, we found that overexpression of WT NPVF, any combination of two distinct RFRP peptides or RFRP2 alone resulted in robust *c-fos* expression along the brain ventricular lining from the hypothalamus to the hindbrain. This ventricular expression pattern is reminiscent of the pattern of phosphorylated ERK that is associated with EGFR-induced rest in rodents (*Gilbert and Davis, 2009*) and *Drosophila* (*Foltenyi et al., 2007*), and is consistent with the link between EGFR and RFamide signaling in promoting sleep in *C. elegans* (*Nath et al., 2016*; *Nelson et al., 2014*) and *Drosophila* (*Lenz et al., 2015*). Among the cells that line the brain ventricle, subependymal neurons secrete into and detect the composition of the CSF (*Vígh et al., 2004*).

These CSF-contacting cells are immunoreactive for neuropeptides found in the CSF, including RFamide neuropeptides (*Chiba et al., 1991*). In humans, several RFamide peptides, including RFRP3, are present in CSF (*Burlet-Schiltz et al., 2002*). Taken together with the *c-fos* expression data, these observations suggest that RFRP peptides may promote sleep by inducing the release of factors into the CSF. This hypothesis is consistent with classical studies that showed the CSF contains sleep-inducing factors (*Krueger et al., 2007*). We note that RFRP2 peptide has yet to be detected *in vivo*, possibly due to the lack of RFRP2-specific antibodies or rapid turnover of the peptide that prevents detection by mass spectrometry, making this study the first demonstration of *in vivo* biological activity of RFRP2. Furthermore, the receptor(s) that bind RFRP2 remain unknown. Taken together, these observations suggest that RFRP2 affects sleep via an unidentified receptor that is expressed in cells along the brain ventricle lining.

Third, our data suggests that NPVF neurons may regulate the processing of sensory information during sleep. This is important because, while a hallmark of sleep is reduced reactivity to sensory stimuli, the underlying mechanism is unclear. Surprisingly, we found that optogenetic stimulation of NPVF neurons is correlated with suppression of neuronal activity in the tectum, a brain region involved in visual processing, to levels similar to those normally observed at night. Consistent with this result, we observed that NPVF neurons extend projections to the tectum (*Figure 5—figure supplement 1A*) suggesting that NPVF neurons may directly regulate neurons in the tectum. Such a mechanism would be surprising because vertebrate sensory systems are thought to be activated similarly during sleep and wake states, with the thalamus limiting responses to these stimuli during sleep (*Coulon et al., 2012*). Our data suggests that sleep-promoting mechanisms may act upstream of the thalamus to inhibit brain regions that receive direct input from sensory systems. Similarly, decreased arousal during sleep stems in part from reduced responsiveness of sensory systems in *C. elegans* (*Cho and Sternberg, 2014*; *Schwarz et al., 2011*).

Finally, we observed that inhibition of NPVF signaling and ablation of NPVF neurons impair both the initiation and maintenance of sleep, hallmark characteristics of insomnia (*Mahowald and Schenck, 2005*). The mechanisms underlying insomnia remain poorly understood, and our data suggests a novel potential mechanism. RFRP peptides are thought to act via GPCRs (*Findeisen et al., 2011*; *Hinuma et al., 2000*), which are highly amenable to small molecule manipulation, and over a quarter of FDA-approved drugs target this protein class (*Overington et al., 2006*). The NPVF system thus provides an attractive therapeutic target for sleep disorders. The development of drugs that target specific sleep regulators is important because most commonly prescribed sleep aids are GABA receptor agonists, and thus are relatively non-specific for sleep, in addition to having significant side effects (*Rihel and Schier, 2013*). The increased efficacy and minimal side effects of recently developed Hcrt receptor antagonists has shown the potential benefits of targeting a more specific sleep regulatory pathway (*Krystal, 2015*). Thus, NPVF receptor antagonists could be useful for treating chronic sleepiness, whereas agonists could be useful to treat insomnia.

In summary, we have shown that NPVF signaling and NPVF neurons comprise a novel vertebrate sleep promoting system. Taken together with roles for RFamide neuropeptides in invertebrate sleep, our results suggest that these peptides participate in an ancient and central sleep regulatory mechanism. Our results redefine the likely functions of vertebrate RFamide neuropeptides and suggest that further comparative studies of RFamide neuropeptides may reveal common principles underlying the regulation of sleep/wake states in the animal kingdom.

## Materials and methods

### Key resources table

| Reagent type or resource | Designation | Source or reference | Identifiers | Additional information |
|---|---|---|---|---|
| gene (*Danio rerio*) | *npvf* | PMID: 26687719 | ZDB-GENE-070424–226 | |
| gene (*Caenorhabditis elegans*) | *flp-13* | PMID: 19910365 | CELE_F33D4.3; WBGene00001456 | |

*Continued on next page*

*Continued*

| Reagent type or resource | Designation | Source or reference | Identifiers | Additional information |
|---|---|---|---|---|
| genetic reagent (*D. rerio*) | *Tg(hs:NPVF)* | this paper | ct846Tg; RRID: ZDB-ALT-170927-2 | zebrafish *npvf* open reading frame cloned downstream of the zebrafish *hsp70c* promoter |
| genetic reagent (*D. rerio*) | *Tg(hs:RFRP1)* | this paper | ct854Tg; RRID: ZDB-ALT-171004-1 | NPVF transgene cloned downstream of the zebrafish *hsp70c* promoter, but with the amino acid sequence of RFRP2 and RFRP3 scrambled to abolish peptide function |
| genetic reagent (*D. rerio*) | *Tg(hsp:RFRP2)* | this paper | ct855Tg; RRID: ZDB-ALT-171004-2 | NPVF transgene cloned downstream of the zebrafish *hsp70c* promoter, but with the amino acid sequence of RFRP1 and RFRP3 scrambled to abolish peptide function |
| genetic reagent (*D. rerio*) | *Tg(hs:RFRP3)* | this paper | ct856Tg; RRID: ZDB-ALT-171004-3 | NPVF transgene cloned downstream of the zebrafish *hsp70c* promoter, but with the amino acid sequence of RFRP1 and RFRP2 scrambled to abolish peptide function |
| genetic reagent (*D. rerio*) | *Tg(hs:RFRP1,2)* | this paper | ct857Tg; RRID: ZDB-ALT-171004-4 | NPVF transgene cloned downstream of the zebrafish *hsp70c* promoter, but with the amino acid sequence of RFRP3 scrambled to abolish peptide function |
| genetic reagent (*D. rerio*) | *Tg(hs:RFRP1,3)* | this paper | ct858Tg; RRID: ZDB-ALT-171004-5 | NPVF transgene cloned downstream of the zebrafish *hsp70c* promoter, but with the amino acid sequence of RFRP2 scrambled to abolish peptide function |
| genetic reagent (*D. rerio*) | *Tg(hs:RFRP2,3)* | this paper | ct859Tg; RRID: ZDB-ALT-171004-6 | NPVF transgene cloned downstream of the zebrafish *hsp70c* promoter, but with the amino acid sequence of RFRP1 scrambled to abolish peptide function |
| genetic reagent (*D. rerio*) | *Tg(hs:RFRPscr)* | this paper | ct860Tg; RRID: ZDB-ALT-171004-7 | NPVF transgene cloned downstream of the zebrafish *hsp70c* promoter, but with the amino acid sequence of RFRP1, RFRP2, and RFRP3 scrambled to abolish peptide function |
| genetic reagent (*D. rerio*) | *Tg(hs:FLP-13)* | this paper | | *C. elegans* flp-13 open reading frame cloned downstream of the zebrafish hsp70c promoter |
| genetic reagent (*D. rerio*) | *npvf* mutant | this paper | ct845; RRID: ZDB-ALT-170927-1 | mutant contains a 7 bp deletion after nucleotide 324 of the open reading frame; premature stop codon after amino acid 146 |
| genetic reagent (*D. rerio*) | *Tg(npvf:EGFP)* | this paper | ct847Tg; RRID: ZDB-ALT-170927-3 | *npvf* promoter cloned upstream of enhanced green fluorescent protein (EGFP) |
| genetic reagent (*D. rerio*) | *Tg(npvf:KalTA4)* | this paper | ct848Tg; RRID: ZDB-ALT-170927-4 | *npvf* promoter cloned upstream of KalTA4 |
| genetic reagent (*D. rerio*) | *Tg(npvf:ReaChR-mCitrine)* | this paper | ct849Tg; RRID: ZDB-ALT-170927-5 | *npvf* promoter cloned upstream of ReaChR-mCitrine |
| genetic reagent (*D. rerio*) | *Tg(npvf:mTagYFP-T2A-eNTR)* | this paper | ct850Tg; RRID: ZDB-ALT-170927-6 | *npvf* promoter cloned upstream of mTagYFP-T2A-enhanced nitroreductase |

*Continued on next page*

*Continued*

| Reagent type or resource | Designation | Source or reference | Identifiers | Additional information |
|---|---|---|---|---|
| genetic reagent (*D. rerio*) | *Tg(UAS:TRPV1-TagRFP-T; cmlc2:EGFP)* | this paper | ct851Tg; RRID: ZDB-ALT-170927-7 | *UAS* promoter cloned upstream of TRPV1-TagRFP-T |
| genetic reagent (*D. rerio*) | *Tg(elavl3:H2B-GCaMP6s)* | PMID: 25068735 | jf5Tg; RRID: ZDB-ALT-141023-2 | |
| antibody | anti-GFP (Chicken polyclonal) | Aves Laboratories | Cat#: GFP-1020; RRID: AB_10000240 | 1:400 |
| antibody | anti-TagRFP (Rabbit polyclonal) | Evrogen | Cat#: AB233 RRID: AB_2571743 | 1:100 |
| antibody | anti-dsRed (Rabbit polyclonal) | Takara Clonetech | Cat#: 645496 | 1:400 |
| antibody | anti-digoxigenin Fab fragments (Sheep polyclonal) | Sigma-Aldrich | Cat#: 11093274910; RRID: AB_514497 | 1:2000 |
| antibody | Alexa 488- or 568-secondaries | Molecular Probes | | 1:500 |
| sequence-based reagent | Primers for genotyping | this paper | | See Materials and Methods |
| sequence-based reagent | Primers for qPCR | this paper | | See Materials and Methods |
| sequence-based reagent | Primers for riboprobe synthesis | this paper | | See Materials and Methods |
| sequence-based reagent | Oligonucleotides used to generate *npvf* CRISPR mutant | this paper | | See Materials and Methods |
| peptide, recombinant protein | RF9 | Tocris | Cat #: 3672 | |
| peptide, recombinant protein | GJ-14 | PMID: 26259035 | Anaspec, Inc; custom synthesis | |
| peptide, recombinant protein | Normal Goat Serum | Thermo Fisher Scientific | Cat# NC9660079 | |
| commercial assay or kit | DIG RNA Labeling Kit | Sigma-Aldrich | Cat#: 11175025910 | |
| commercial assay or kit | TSA Plus Cyanine 3 and Fluorescein System | PerkinElmer | Cat#: NEL753001KT | |
| commercial assay or kit | FirstChoice RLM-RACE | Thermo Fisher Scientific | Cat#: AM1700 | |
| commercial assay or kit | SuperScript III First-Strand Synthesis System | Thermo Fisher Scientific | Cat#: 18080051 | |
| chemical compound, drug | 16% paraformaldehyde | Thermo Fisher Scientific | Cat#: 15710 | |
| chemical compound, drug | NBT/BCIP Stock Solution | Thermo Fisher Scientific | Cat#: 11681451001 | |
| chemical compound, drug | SYBR Green PCR Master Mix | Thermo Fisher Scientific | Cat#: 4364346 | |
| software, algorithm | Excel | Microsoft | https://office.microsoft.com/excel/ | |
| software, algorithm | Fiji | PMID: 22732772 | https://fiji.sc; RRID: SCR_002285 | |
| software, algorithm | GraphPad Prism6 | GraphPad Software | http://www.graphpad.com/; RRID: SCR_002798 | |
| software, algorithm | MATLAB | MathWorks | https://www.mathworks.com/ | |

*Continued on next page*

*Continued*

| Reagent type or resource | Designation | Source or reference | Identifiers | Additional information |
|---|---|---|---|---|
| software, algorithm | Principal Component and Independent Component Analysis | PMID: 19778505 | | Automated analysis of cellular signals from large-scale calcium imaging data. |
| software, algorithm | Videotracking script for behavioral analysis | PMID: 17182791 | | Source code attached |
| software, algorithm | Arousal threshold script for behavioral analysis | PMID: 26374985 | | Source code attached |
| other | 96-well plate for behavioral experiments | GE Healthcare Life Sciences | Cat#: 7701–1651 | |
| other | MicroAmp Optical Adhesive Film | Thermo Fisher Scientific | Cat#: 4311971 | |

## Zebrafish genetics

Zebrafish were raised on a 14:10 h light:dark cycle at 28.5°C, with lights on at 9 a.m. and off at 11 p.m., unless specified otherwise. Wild-type (WT), transgenic, and mutant stocks come from a background of TL X AB strains. All experiments were performed using standard protocols (*Westerfield, 1993*) in accordance with the California Institute of Technology and University of Southern California Institutional Animal Care and Use Committee guidelines.

## Generation of transgenic lines

### Tg(hs:NPVF)

Full-length zebrafish *npvf* cDNA was isolated using 5' and 3' RACE (FirstChoice RLM-RACE, Ambion) and the open reading frame was cloned downstream of the zebrafish *hsp70c* promoter (*Halloran et al., 2000*) in a vector containing flanking I-SceI meganuclease sites. We generated stable transgenic animals by injecting plasmids with I-SceI (New England Biolabs Inc.) into the cell of embryos at the one-cell stage (*Thermes et al., 2002*). To identify transgenic founders, we outcrossed potential founders, performed a heat shock on their larval progeny at 5 dpf, and then performed *npvf* ISH. At least three lines that produced strong and ubiquitous overexpression, and lacked pre-heat shock expression, were selected and tested for behavioral phenotypes. *Tg(hs:NPVF)* transgenic animals (ct846Tg, RRID: ZDB-ALT-170927-2) were identified by PCR using the primers 5'-GCACACCTGAATCACCATCA-3' and 5'-GGTTTGTCCAAACTCATCAATGT-3' with a product size of 206 bp.

### Heat-shock inducible *npvf* transgenes containing scrambled RFRP peptides

We identified RFRP peptides based on homology to the mammalian (*Ubuka and Tsutsui, 2014*) and goldfish (*Sawada et al., 2002*) peptides. We generated transgenes in which the amino acid sequence of one or more peptide was scrambled to abolish peptide function. We verified that scrambled peptides were distinct from endogenous zebrafish peptides using BLASTp (*Altschul et al., 1997*) and selected codons based on codon usage frequencies in the zebrafish genome (*Horstick et al., 2015*). Transgenes were assembled using double-stranded DNA gBlocks (Integrated DNA Technologies) and Gibson assembly (*Gibson et al., 2009*). Stable lines for *Tg(hs:RFRP1)* (ct854Tg, RRID: ZDB-ALT-171004-1), *Tg(hs:RFRP2)* (ct855Tg, RRID: ZDB-ALT-171004-2), *Tg(hs:RFRP3)* (ct856Tg, RRID: ZDB-ALT-171004-3), *Tg(hs:RFRP1,2)* (ct857Tg, RRID: ZDB-ALT-171004-4), *Tg(hs:RFRP1,3)* (ct858Tg, RRID: ZDB-ALT-171004-5), *Tg(hs:RFRP2,3)* (ct859Tg, RRID: ZDB-ALT-171004-6), and *Tg(hs:RFRPscr)* (ct860Tg, RRID: ZDB-ALT-171004-7) were generated and isolated as described for *Tg(hs:NPVF)*. The presence of any *npvf* transgene was confirmed by PCR using the NPVF primers 5'-GCACACCTGAATCACCATCA-3' and 5'-GGTTTGTCCAAACTCATCAATGT-3' with a product size of 206 bp. Fish containing transgenes with specific scrambled RFRP peptides were identified by PCR using the primers: RPRP1 scrambled: 5'-CTTCGCTCTTCTTTCTTTAGCC-3' and 5'-AGTGAAGTGGAGCGTGCAAT-3' (266 bp product); RFRP2 scrambled: 5'-CTTCGCTCTTCTTTCTTTAGCC-3' and 5'-GATCTTCGTGGACGCGTCGAA-3' (349 bp product); RFRP3 scrambled: 5'-C

TTCGCTCTTCTTTCTTTAGCC-3' and 5'-TGCAGGGGTTGAGAACTGA-3' (415 bp product). As an example, a *Tg(hs:RFRP2)* animal would produce PCR products using NPVF primers (206 bp), RFRP1 scrambled primers (266 bp), and RFRP3 scrambled primers (415 bp), and would not produce a PCR product using RFRP2 scrambled primers.

## Tg(hs:FLP-13)

The *C. elegans flp-13* open reading frame was amplified from a cDNA library from the *C. elegans* strain PS6845, and cloned downstream of the zebrafish *hsp70c* promoter (*Halloran et al., 2000*) in a vector containing flanking I-SceI meganuclease sites. We generated stable transgenic animals by injecting plasmid with I-SceI (New England Biolabs, Inc.) into the cell of embryos at the one-cell stage (*Thermes et al., 2002*). To identify transgenic founders, we outcrossed potential founders, performed a heat shock on their larval progeny at 5 dpf, and then performed ISH using a *flp-13* specific probe. A line that produced strong and ubiquitous overexpression in the brain was used for behavioral phenotypes. Transgenic animals were identified by PCR using the primers 5'-ATGA TGACGTCACTGCTCACTATC−3' and 5'-GGTTTGTCCAAACTCATCAATGT−3' with a product size of 554 bp.

## npvf mutant

The *npvf* mutant (ct845Tg, RRID: ZDB-ALT-170927-1) was generated using CRISPR/Cas9 as described (*Hwang et al., 2013*), with the sgRNA target sequence 5'-GGGAGGTTGATGGTAGACTT-3'. The mutant contains a 7 bp deletion (CCCAAGT) after nucleotide 324 of the open reading frame. The mutation results in a change in reading frame after amino acid 108 and a premature stop codon after amino acid 146, compared to 198 amino acids for the WT protein. The predicted mutant protein lacks the RFRP2 and RFRP3 peptide sequences. Mutant animals were genotyped using the primers 5'-CAGTGGTGGTGCGAGTTCT-3' and 5'-GCTGAGGGAGGTTGATGGTA-3', which produce a 151 bp or 144 bp band for the WT or mutant allele, respectively. *npvf* heterozygous mutants were outcrossed to the parental TLAB strain for three generations before use in behavioral experiments. *npvf* mutants are homozygous viable and fertile, and are morphologically indistinguishable from WT animals.

## Tg(npvf:EGFP) and Tg(npvf:KalTA4)

A 3.9 kb region of genomic DNA immediately 5' to the *npvf* start codon was amplified from zebrafish genomic DNA using the primers 5'-TGACTGAAGTAGAAAATCAGCCTTT-3' and 5'-CTTACAA TCGGTCACTGAAGGC-3' and Pfu Ultra II Fusion HS DNA Polymerase (Agilent Technologies). This sequence was subcloned downstream of 2 copies of a neuron-restrictive silencing element (NRSE) (*Bergeron et al., 2012*; *Xie et al., 2012*) and upstream of enhanced green fluorescent protein (EGFP) or KalTA4 (*Distel et al., 2009*) in a vector containing flanking I-SceI meganuclease sites using Gibson assembly to generate *Tg(npvf:EGFP)* (ct847Tg, RRID: ZDB-ALT-170927-3) and *Tg(npvf: KalTA4)* (ct848Tg,RRID:ZDB-ALT-170927-4) animals. Stable transgenic lines were generated using the I-SceI method (*Thermes et al., 2002*). *Tg(npvf:KalTA4)* animals were identified by PCR using the primers 5'-ATGCAAAGCTGTGAGTGCAT-3' and 5'-TTGTGAGTGGACTTCGCTTG-3' (270 bp product).

## Tg(npvf:ReaChR-mCitrine) and Tg(npvf:mTagYFP-T2A-eNTR)

Using Gibson assembly (*Gibson et al., 2009*), the 3.9 kb *npvf* promoter was cloned upstream of ReaChR (*Lin et al., 2013*) fused to mCitrine, and upstream of mTagYFP-T2A-enhanced nitroreductase (eNTR) (*Mathias et al., 2014*; *Tabor et al., 2014*), in a vector containing NSRE elements and flanking I-SceI meganuclease sites. The *npvf:mTagYFP-T2A-eNTR* transgene contains a Gap43 palmitoylation sequence that targets TagYFP to the membrane, and a T2A sequence, which generates a self-cleaving peptide (*Donnelly et al., 2001*), was inserted between mTagYFP and eNTR to produce stoichiometric levels of each protein. Stable transgenic lines were generated using the I-SceI method (*Thermes et al., 2002*). Transgenic animals were identified using fluorescence or by PCR using the primers: ReaChR: 5'-CACGAGAGAATGCTGTTCCA-3' and 5'-CCATGGTGCGTTTGCTATAA-3' (450 bp product); eNTR: 5'-ATGCAAAGCTGTGAGTGCAT-3' and 5'-CTCGCCTTTGCTAACCATTG-3' (227 bp product). For simplicity, *Tg(npvf:ReaChR-mCitrine)* (ct849Tg, RRID: ZDB-ALT-170927-5) is

referred to as *Tg(npvf:ReaChR)*, and *Tg(npvf:mTagYFP-T2A-eNTR)* (ct850Tg, RRID: ZDB-ALT-170927-6) is referred to as *Tg(npvf:eNTR)*, in the figures.

### Tg(UAS:TRPV1-TagRFP-T;cmlc2:EGFP)

We generated the *Tg(UAS:TRPV1-TagRFP-T;cmlc2:EGFP)* transgenic line (ct851Tg, RRID: ZDB-ALT-170927-7) using the Tol2kit (*Kwan et al., 2007*) and multisite Gateway Technology (Invitrogen). The middle entry clone (pME-TRPV1-TagRFP-T) was generated using a BP reaction by combining a donor vector (pDONR221) and TRPV1-TagRFP-T (*Chen et al., 2016c*). The LR reaction was performed by combining the 5' (p5E-UAS), middle (pME-TRPV1-TagRFP-T), and 3' (p3E-polyA) entry clones into the destination vector (pDestTol2CG2). Stable transgenic lines were generated by injecting the plasmid and *tol2 transposase* mRNA into embryos at the 1 cell stage. Transgenic animals were identified by EGFP expression in the heart or by PCR using the primers 5'-CAGCCTCACTTTGAGCTCCT-3' and 5'-TCCTCATAAGGGCAGTCCAG-3' (349 bp product). For simplicity, *Tg(UAS:TRPV1-TagRFP-T;cmlc2:EGFP)* is referred to as *Tg(UAS:TRPV1)* in the figures.

*Tg(elavl3:H2B-GCaMP6s)*; *nacre* fish were a kind gift from Misha Ahrens (*Vladimirov et al., 2014*).

## Zebrafish behavioral assays

### Locomotor activity assay

Individual larvae were placed into each well of a 96-well plate (7701–1651, Whatman) containing 650 µL of E3 embryo medium (5 mM NaCl, 0.17 mM KCl, 0.33 mM CaCl$_2$, 0.33 mM MgSO$_4$, pH 7.4). The only exception was the chemogenetic ablation experiment, in which animals were placed into the 96-well plate after metronidazole treatment. Plates were sealed with an optical adhesive film (4311971, Applied Biosystems) to prevent evaporation. The sealing process introduces air bubbles in some wells, which are discarded from analysis. Animals were blindly assigned a position in the plate and were genotyped by PCR after the behavioral experiment was complete. The only exception to this setup was the experiment in which +/+, *Tg(hs:RFRP)/+*, and *Tg(hs:RFRP)/Tg(hs:RFRP)* animals were compared. Because we could not distinguish between heterozygous and homozygous transgenic animals using the PCR genotyping assay, homozygous *Tg(hs:RFRP)* cousins were placed into interleaving rows of a plate that also contained blindly assigned +/+ and *Tg(hs:RFRP)/+* siblings, which were identified by PCR following the experiment. Locomotor activity was monitored using an automated videotracking system (Viewpoint Life Sciences) with a Dinion one-third inch monochrome camera (Dragonfly 2, Point Grey) fitted with a fixed-angle megapixel lens (M5018-MP, Computar) and infrared filter. For heat shock-induced overexpression experiments, larvae were heat shocked at 37°C for 1 hr starting at 12 p.m., 3 p.m., 4 p.m., or 9:45 p.m. at 5 dpf. The movement of each larva was captured at 15 Hz and recorded using the quantization mode with 1 min time bins. The 96-well plate and camera were housed inside a custom-modified, Zebrabox (Viewpoint Life Sciences) that was continuously illuminated with infrared LEDs, and illuminated with white LEDs from 9 a.m. to 11 p.m., except as noted in LL or DD experiments. The 96-well plate was housed in a chamber filled with recirculating water to maintain a constant temperature of 28.5°C. The parameters used for detection were: detection threshold, 15; burst, 29; freeze, 3, which were determined empirically. Data were processed using custom PERL and Matlab (The Mathworks, Inc.) scripts, and statistical tests were performed using Prism (GraphPad) for ANOVA analysis and Excel (Microsoft) for two-tailed Student's *t*-test.

A movement was defined as a pixel displacement between adjacent video frames preceded and followed by a period of inactivity of at least 67 ms (the limit of temporal resolution). Any one-minute period with no movement was defined as one minute of sleep based on arousal threshold changes (*Prober et al., 2006*). A sleep bout was defined as a continuous string of sleep minutes. Sleep latency was defined as the length of time from lights on or off to the start of the first sleep bout. Average activity was defined as the average amount of activity in seconds/hour, including sleep bouts. Average wake activity was defined as the average amount of activity in seconds/hour, excluding sleep bouts.

## Arousal threshold assay

We modified the videotracking system by adding an Arduino-based automated driver to control two solenoids (28P-I-12, Guardian Electric) that delivered a tap to a 96-well plate containing larvae (*Singh et al., 2015*). This setup allowed us to drive the solenoids with voltage ranging from 0 V to 20 V over a range of 4095 settings (from 0.01 to 40.95). We used taps ranging from a power setting of 1–36.31. Taps of 14 different intensities were applied in a random order with an inter-trial-interval of 1 min during the day and night for *Tg(hs:NPVF)* and RF9/GJ-14 experiments, respectively. Previous studies showed that a 15 s interval between repetitive stimuli is sufficient to prevent behavioral habituation (*Burgess and Granato, 2007*; *Woods et al., 2014*). The background probability of movement was calculated by identifying for each genotype the fraction of larvae that moved 5 s prior to all stimuli delivered during an experiment (14 different tap powers x 30 trials per experiment = 420 data points per larva). This value was subtracted from the average response fraction value for each tap event. The response of larvae to the stimuli was monitored using the videotracking software and was analyzed using Matlab and Excel. Statistical analysis was performed using the Variable Slope log(dose) response curve fitting module of Prism.

## Optogenetic behavioral assay

The videotracking system was modified to include a custom array containing blue LEDs (470 nm, MR-B0040-10S, Luxeon V-star) mounted 15 cm above and 7 cm away from the center of the 96-well plate to ensure uniform illumination (*Singh et al., 2015*). The LEDs were controlled using a custom-built driver and software written in BASIC stamp editor. A power meter (1098293, Laser-check) was used before each experiment to verify uniform blue light intensity (~400 μW at the surface of the 96-well plate). During the afternoon of 5 dpf, single larvae were placed into each well of a 96-well plate as described above and placed in the videotracker in the dark for 7 h. Larvae were then exposed to blue light for 30 min, starting at 12 a.m. Three trials were performed during the night, with an inter-trial interval of 3 hr. Total activity for each larva was monitored for 30 min before and after light onset, with data collected in 10 s bins. Light onset caused a burst of locomotor activity lasting for ~30 s for all genotypes, so data obtained during the minute before and after light onset was excluded from analysis. A large burst of locomotor activity was also observed for all genotypes when the lights were turned off after the 30 min illumination period. This data was excluded from analysis and is not shown in the figures. The total amount of locomotor activity of each larva during the 30 min of light exposure, excluding the minute after light onset, was divided by the average baseline locomotor activity for all larvae of the same genotype. The baseline period was defined as 30 min before light onset, excluding the minute before light onset. Data from three independent experiments were pooled and converted to percentage of WT larvae.

## Chemogenetic behavioral assay

Neuronal activation using TRPV1 was performed as described (*Chen et al., 2016c*) with some modifications. Animals generated by mating homozygous *Tg(npvf:KalTA4)* to heterozygous *Tg(UAS: TRPV1-TagRFP-T)* fish were immersed in either DMSO (4948–02, Macron Chemicals) vehicle control or 2 μM capsaicin at 100 h post-fertilization (hpf). Capsaicin powder (M2028, Sigma) was dissolved in DMSO to prepare a 100 mM stock solution that was stored in aliquots at −20°C. Capsaicin working solutions were prepared just before each experiment by diluting the stock solution in E3 medium. All treatments contained a final concentration of 0.002% DMSO. Behavioral analysis was performed from 6 dpf until 8 dpf. Larvae were then genotyped by PCR to identify *Tg(npvf:KalTA4); Tg(UAS: TRPV1-TagRFP-T)* and *Tg(npvf:KalTA4)* animals.

## Metronidazole (MTZ) induced neuronal ablation

Neuronal ablation using eNTR was performed as described (*Gandhi et al., 2015*) with some modifications. Heterozygous fish from a *Tg(npvf:mTagYFP-T2A-eNTR)* line that exhibits strong YFP fluorescence in NPVF neurons were outcrossed to WT fish. Embryos were raised in E3 medium until 60 hpf, at which point they were treated with 5 mM MTZ (46461, Sigma Aldrich) in 0.2% DMSO in E3 medium for 60 h (60–120 hpf), with the MTZ solution refreshed every ~20 h. Larvae were then rinsed twice with E3 medium (120–122 hpf) and maintained in E3 medium for ~8 hr. MTZ treated *Tg(npvf: mTagYFP-T2A-eNTR)* and WT siblings were then placed into 96-well plates and allowed to recover

from MTZ treatment for 24 h. Behavioral analysis was performed from 6 dpf at night until 8 dpf at night. Larvae were then genotyped by PCR to identify transgenic and WT animals. To quantify MTZ-induced neuronal ablation, *Tg(npvf:mTagYFP-T2A-eNTR)* and WT siblings were fluorescently sorted at 2 dpf, treated with either MTZ or DMSO vehicle as described above, and then processed at 7 dpf for ISH using a DIG-labeled *npvf*-specific probe as described below.

## Scripts used for analysis of behavioral data

*sort_fish_sttime_192.pl* is a Perl script (**Prober et al., 2006**) that converts data acquired by the View-point videotracker system to a format that is useful for analysis using Matlab and removes notations that are not relevant to behavioral analysis. *perl_batch_192well.m* is a Matlab script that allows the *sort_fish_sttime_192.pl* script to run on the Matlab platform. *TapAnalysis.m* is a Matlab script that analyzes tapping assay data and generates a table that lists the number of larvae that moved during each tapping event. *VT_analysis.m* is a Matlab script (modified from [**Prober et al., 2006**]) that analyzes data collected by the Viewpoint videotracker system to quantify several metrics, including locomotor activity, wake activity, sleep, sleep architecture and sleep latency. These scripts and detailed instructions on their use will be provided upon request.

## Histology

### *In situ* hybridization (ISH)

Samples were fixed in 4% paraformaldehyde (PFA) in phosphate buffered saline (PBS) for approximately 16 h at room temperature. ISH was performed using digoxygenin (DIG) labeled antisense riboprobes (DIG RNA Labeling Kit, Roche) as previously described (**Thisse and Thisse, 2008**). Double-fluorescent ISH (FISH) was performed using DIG- and fluorescein-labeled riboprobes and the TSA Plus Fluorescein and Cyanine 3 Systems kit (Perkin Elmer). Probes specific for *npvf* (**Tessmar-Raible et al., 2007**), *vesicular glutamate transporter 2a* (*vglut2a*), *vglut2b*, and *glutamate decarboxylase 67* (*gad67*) (**Higashijima et al., 2004**) have been described. The *c-fos* probe was transcribed using a PCR product amplified from a zebrafish cDNA library using the primers Forward: 5'-CAGC TCCACCACAGTGAAGA-3' and Reverse: 5'-TGCAAACAATTCGCAAGTTC-3', and then serially amplified with the same forward primer and a T7 sequence added to the Reverse Primer: 5'-GAA TTGTAATACGACTCACTATAGGGTGCAAACAATTCGCAAGTTC-3'.

### Immunohistochemistry

Samples were fixed in 4% PFA in PBS overnight at 4°C and then washed with 0.25% Triton X-100/PBS (PBTx). Brains were manually dissected and blocked for at least 1 h in 2% goat serum/2% DMSO/PBTx at room temperature or overnight at 4°C. Antibody incubations were performed in blocking solution overnight at 4°C using chicken anti-GFP (1:400, GFP-1020, Aves Labs), rabbit anti-TagRFP (1:100, AB233, Evrogen), or rabbit anti-dsRed (1:400, 643496, Takara Clontech) primary antibodies, and Alexa Fluor 488 and 568 secondary antibodies (1:500, Life Technologies). Samples were mounted in 50% glycerol/PBS and imaged using a Zeiss LSM 780 confocal microscope. Quantification of neurons (*Figure 5—figure supplement 2M* and *Figure 6C*) and *c-fos* intensity per NPVF neuron (*Figure 5—figure supplement 2G*) was performed blind prior to genotyping.

## Pharmacology

RF9 (3672; Tocris) was dissolved in DMSO and then added to E3 medium for a final concentration of 0.1% DMSO and 10 μM RF9. Drug solution was freshly prepared prior to each experiment. Controls were exposed to 0.1% DMSO alone. RF9 and DMSO vehicle were loaded into separate sides of a 96-well plate with an empty row in between. WT larvae from the same clutch were added to the plate, and plates were sealed with an optical adhesive film (4311971, Applied Biosystems) to prevent evaporation.

GJ-14 (Diphenylacetamide-D-Arg-D-Phg-NH$_2$) was custom synthesized by AnaSpec, Inc. (Fremont, CA) as previously described (**Kim et al., 2015**). Residual TFA was removed and exchanged with acetate salt. Purity was verified by HPLC, found to be 99% around the peak area, and subsequently lyophilized. GJ-14 was reconstituted in DMSO, and then added to E3 medium for a final concentration of 0.05% DMSO and 10 μM GJ-14. Drug solution was freshly prepared prior to each experiment and behavioral experiments were performed as described for RF9.

## qPCR

Larval zebrafish were raised on a 14:10 h light:dark cycle at 28.5°C with lights on at 9 a.m. and off at 11 p.m. At 5 dpf, total RNA was collected using Trizol reagent (15596–026, Life Technologies) from 20 pooled larvae every 6 h for 36 h. cDNA was synthesized from 5 µg of total RNA using Superscript III Reverse Transcriptase (18080–051, Invitrogen) and quantitative PCR was carried out using SYBR green master mix (4364346, Life Technologies) in an ABI PRISM 7900HT (Life Technologies) instrument using the primers 5'-GGCTCTCAGATTGCCACTTT-3' and 5'-GGGGCCACATTAAGAGTGAA-3'. *ribosomal protein l13a* (*rpl13a*) was used as a reference gene, using the primers 5'-TCTGGAG-GACTGTAAGAGGTATGC-3' and 5'-AGACGCACAATCTTGAGAGCAG-3'. Relative expression levels were determined by using the ΔΔCt method (*Livak and Schmittgen, 2001*), normalized to the highest Ct value for each gene.

## Imaging

### Confocal and brightfield imaging

Dissected brains were coverslip mounted in Vectashield (H-1000, Vector Labs) or 80% glycerol in PBS and imaged using a compound microscope (Axioimager with EC Plan-Neofluar 10x/0.30 NA air objective or Plan-Apochromat 20x/0.8 NA air objective, Zeiss) for chromogenic ISH samples, or for double fluorescent ISH samples, a confocal microscope (LSM 780 with Plan-Apochromat 10x/0.45 NA air objective, LD LCI Plan-Apochromat 25x/0.8 NA Imm Corr objective, or LD C-Apochromat 40x/1.1 NA water objective, Zeiss). Fluorescein and cyanine were imaged in separate channels with 488 nm and 561 nm lasers, respectively. Confocal images are displayed as the maximum intensity z-projection of a stack of optical sections of approximately one airy unit (A.U.) thickness.

### Two-photon selective plane illumination microscopy (2P-SPIM)

Live imaging was performed using a custom built 2P-SPIM microscope (*Truong et al., 2011*) with 940 nm excitation pulsed laser light (Chameleon Ultra 2, Coherent), focused by spherical optics (Nikon MRH07120, 10x, NA = 0.3), yielding a focused beam waist of approximately 4 µm, with total power of 50 mW. Fluorescence signal was collected by a water-immersion objective (Nikon MRD77220, 25x, NA = 1.1 for data shown in *Figure 7* and *Figure 7—figure supplement 1*, Olympus XLUMPLFLN-W, 20x, NA = 1.0 for data shown in *Figure 7—figure supplement 2*), sCMOS camera (Hamamatsu ORCA-Flash 4.2), and appropriate spectral filters (Semrock). To increase the signal-to-noise ratio, we used a tube lens of focal length = 100 mm (Thorlabs AC508-1000-A-NK) to yield a 2x de-magnification. Image acquisition was performed using MicroManager software (*Edelstein et al., 2014*), with 2x binning, 950 ms exposure time.

Larvae were raised in a light:dark cycle, with dim white lights on from 9 a.m. to 10 p.m., for 4–7 days in order to entrain circadian rhythms. At ~20 hpf, 1-phenyl-2-thiourea (PTU) (30 mg/L) was added to reduce pigmentation. One h prior to imaging, larvae were anesthetized with tricaine (100 mg/L) and embedded in 1.5% low-melting point agarose (SeaPlaque), then mounted in the imaging chamber filled with 35 mL of 0.3x Danieau Buffer (1740 mM NaCl, 21 mM KCl, 12 mM MgSO$_4$•7H$_2$O, 18 mM Ca(NO$_3$)$_2$, 150 mM HEPES) at 28°C without anesthetic, and allowed to recover for at least 30 min. Live imaging was performed with the focal plane set at depth of ~70 µm from the dorsal surface of the brain.

For results presented in *Figure 7A–D*, *Tg(elavl3:H2B-GCaMP6s)* animals were entrained for 4.5 days, mounted for imaging at 5 dpf in the evening, and then imaged in 2 min sessions every 15 min for up to 36 h in constant dark. Intrinsic activity and light-evoked responses were used to assess putative sleep or wake states. Each 2 min session consisted of three stages that recorded: (1) intrinsic neural activity for 30 s, after which (2) a 625 nm LED (Thorlabs M625L3) was turned on and slowly increased in intensity for 60 s, achieving a final intensity of 0.1 µW/cm$^2$ at the sample, after which (3) the LED was abruptly turned off and imaging continued for an additional 30 s. Using this approach, we recorded both intrinsic activity and activity evoked by onset and offset of an arousing light stimulus.

For results presented in *Figure 7E–H* and *Figure 7—figure supplement 1*, in order to test the hypothesis that stimulation of NPVF neurons induces neuronal activity that is similar to that observed during subjective night, we optogenetically stimulated NPVF neurons using *Tg(npvf:ReaChR-mCitrine);Tg(elavl3:H2B-GCaMP6s);nacre/nacre* larvae, with *Tg(elavl3:H2B-GCaMP6s);nacre/nacre*

siblings as controls. Animals were entrained up to the day of imaging, at 5 or 6 dpf. Imaging was performed in 2 min sessions every 5 min using the same settings and region of interest as for recording brain activity during natural sleep and wake. Baseline neural activity was established in five imaging trials (25 min). A ReaChR-activating 625 nm LED (Thorlabs M625L3) was then turned on and another 10 imaging trials were performed. The activating light had intensity of 10 mW/cm$^2$, slightly focused with beam waist of ~2 mm at the sample, covering the entire head of the animal. Imaging data was collected blinded to genotype, and posthumous PCR genotyping was performed as described above.

Image analysis for *Figure 7* and *Figure 7—figure supplement 1* was performed using ImageJ (NIH). Background signal due to camera dark counts and residual light from the 625 nm LED was subtracted. For all quantitative analyses, the fluorescence signal was averaged over a region of interest of the optic tectum (*Figure 7A*). Fluorescent signal intensity is plotted as F in *Figure 7A* and as $\Delta F = F - F_0$ in *Figure 7B and D*, where the baseline $F_0$ = lowest signal measured across the entire subjective day/subjective night recording period. In *Figure 7C*, signal is normalized by $F_N$ = averaged signal of intrinsic activity during subjective day. In *Figure 7E–H*, the signal for each animal is calculated with baseline $F_0$ = averaged signal over the first five trials, then normalized by $F_N$ = peak of signal after red light is turned on, for the same animal. The steady-state value shown in *Figure 7H* was calculated by averaging the normalized signal over the last six trials (trials 10–15).

For results presented in *Figure 7—figure supplement 2*, in order to test the hypothesis that chemogenetic stimulation of NPVF neurons affects neuronal activity in a manner expected for increased sleep at night, as expected based on results from the behavioral assay (*Figure 5—figure supplement 2*), we chemogenetically stimulated NPVF neurons in *Tg(npvf:KalTA4);Tg(UAS:TRPV1-TagRFP-T);Tg(elavl3:H2B-GCaMP6s);nacre/nacre* larvae and their *Tg(npvf:KalTA4);Tg(elavl3:H2B-GCaMP6s); nacre/nacre* control siblings. We entrained the animals as described above, and starting at 4 dpf we treated the animals with 2 µM Csn in rearing media. We then mounted the animals for imaging at 6 dpf in the afternoon. The mounted animals were maintained in the same light:dark cycle as during entrainment, and spontaneous calcium activity was recorded in 2 min sessions at 1 Hz, repeated every 15 min, for 20–24 hr across the night of 6 dpf. Imaging data was collected blinded to genotype, and posthumous PCR genotyping was performed as described above.

Image analysis for *Figure 7—figure supplement 2* was performed using Matlab. After background signal was subtracted, we performed standard Principal Component and Independent Component Analysis (PCA/ICA) (*Mukamel et al., 2009*) to automatically segment neurons that exhibited spiking activity. We typically utilized the first 15 principal components (covering ~75% of the total signal variance), as higher order components started to be noise-dominated (determined by visual inspection of components plotted onto the raw data time series). From raw images, we found that neuronal nuclei have nominal diameter = 5 µm, hence ICA was performed with spatial filter of that size. Typically, 500–800 active neurons were extracted from each 2 min imaging session. Some false positives were extracted from skin auto-fluorescence, contributing about 5% of the total segmented neurons, which did not contribute significantly to subsequent quantitative analysis. To establish the baseline signal $F_0$ for each segmented neuron, the temporal signal trace for each 2 min imaging trial was smoothed with a sliding window of 5 s, then the minimum of the resulting signal trace was set to be $F_0$. $\Delta F = F - F_0$ was then calculated for each segmented neuron, for each time step. *Figure 7—figure supplement 2C and D* show $\Delta F$ during the day and night, averaged over all segmented neurons. In *Figure 7—figure supplement 2E, F and G*, day activity values were calculated as the average of both day periods (before and after the night). In *Figure 7—figure supplement 2E*, $\Delta F/F_N$ is shown during the day and night, averaged over all segmented neurons, where $F_N$ = average day activity of the same animal. In *Figure 7—figure supplement 2F*, we compare the non-normalized day neural activity, in units of fluorescence counts, and found that day neural activity was not statistically different between *Tg(UAS:TRPV1-TagRFP-T)* and *Tg(npvf:KalTA4);Tg(UAS:TRPV1-TagRFP-T)* animals. Thus, in order to compare neural activity between the two genotypes at night, we normalized night activity by $F_N$ = average day activity of the same animal (*Figure 7—figure supplement 2G*). Image rendering was performed using ImageJ. Graphs were generated and statistical analysis was performed using MATLAB.

## Statistical analysis

Line graphs in *Figures 1–3* and *6*, *Figure 1—figure supplements 1–3*, *Figure 2—figure supplements 1–2*, *Figure 3—figure supplement 2*, *Figure 5—figure supplement 2* were generated from raw data and smoothed over 1 hr bins in 10 min intervals to show underlying behavioral trends. Line and bar graphs show mean ± standard error of the mean (SEM). In all statistical tests, the significance threshold was set to $p < 0.05$, and *P* values were adjusted for multiple comparisons where appropriate. Parametric analyses were applied because the data followed an approximately normal distribution. Unpaired two-tailed Student's *t*-test was performed using Excel. One-way and Two-way analysis of variance (ANOVA) and post hoc tests to correct for multiple comparisons were performed using Prism. The Holm-Sidak post hoc test was used to correct for multiple comparisons and to allow pairwise comparison of means for all samples. In *Figure 2*, Dunnett's post hoc test was used to compare each of a number of treatments (RFRP transgene variants) with a single control (transgenic animal where all RFRP peptides were scrambled). In *Figure 7H* and *Figure 7—figure supplement 2F and G*, appropriateness of a *t*-test was confirmed by the Kolmogorov-Smirnov normality test ($p > 0.8$). Then, either a one- or two-sampled *t*-test was carried out as described. Asterisks in figures denote statistics for pairwise or multiple comparisons as indicated.

## Acknowledgements

We thank members of the Prober Lab, Paul Sternberg, Elly Chow, Ravi Nath and Han Wang for discussions, and Jason Rihel, Seth Blackshaw and Chanpreet Singh for manuscript comments, as well as Daisy Chilin, Tasha Cammidge, and Hannah Hurley for technical assistance. This work was supported by grants from the NIH (DAL: K99NS097683, F32NS084769; GO: F32NS082010; SEF: MH107238; DAP: NS070911, NS101158, NS095824 and NS101665); the Moore Foundation (SEF); and the Mallinckrodt (DAP), Rita Allen (DAP) and Brain and Behavior Research Foundations (DAP, DAL). We declare no competing interests.

## Additional information

### Funding

| Funder | Grant reference number | Author |
| --- | --- | --- |
| National Institutes of Health | F32NS084769 | Daniel A Lee |
| National Institutes of Health | K99NS097683 | Daniel A Lee |
| Brain and Behavior Research Foundation | 25392 | Daniel A Lee |
| National Institutes of Health | F32NS082010 | Grigorios Oikonomou |
| Gordon and Betty Moore Foundation | | Scott E Fraser |
| National Institutes of Health | MH107238 | Scott E Fraser |
| Edward Mallinckrodt, Jr Foundation | | David A Prober |
| Rita Allen Foundation | | David A Prober |
| Brain and Behavior Research Foundation | | David A Prober |
| National Institutes of Health | NS070911 | David A Prober |
| National Institutes of Health | NS101665 | David A Prober |
| National Institutes of Health | NS101158 | David A Prober |
| National Institutes of Health | NS095824 | David A Prober |

The funders had no role in study design, data collection and interpretation, or the decision to submit the work for publication.

## Author contributions

Daniel A Lee, Conceptualization, Resources, Data curation, Formal analysis, Supervision, Funding acquisition, Validation, Investigation, Visualization, Methodology, Writing—original draft, Project administration, Writing—review and editing; Andrey Andreev, Thai V Truong, Resources, Data curation, Formal analysis, Investigation, Methodology; Audrey Chen, Formal analysis, Investigation, Methodology; Andrew J Hill, Viveca Sapin, Resources, Investigation, Methodology; Grigorios Oikonomou, Resources, Software, Formal analysis, Methodology, Writing—review and editing; Uyen Pham, Laura Glass, Jae Engle, Resources, Methodology; Young K Hong, Formal analysis, Investigation; Steven Tran, Resources; Scott E Fraser, Conceptualization, Resources, Supervision, Funding acquisition, Writing—review and editing; David A Prober, Conceptualization, Resources, Supervision, Funding acquisition, Investigation, Writing—original draft, Project administration, Writing—review and editing

## Author ORCIDs

Daniel A Lee  http://orcid.org/0000-0001-7411-2740
Steven Tran  http://orcid.org/0000-0001-8515-8250
David A Prober  http://orcid.org/0000-0002-7371-4675

## Ethics

Animal experimentation: This study was performed in strict accordance with the recommendations in the Guide for the Care and Use of Laboratory Animals of the National Institutes of Health. All experiments were performed using standard protocols (Westerfield, 1993) in accordance with the California Institute of Technology and University of Southern California Institutional Animal Care and Use Committee guidelines.

## Decision letter and Author response

Decision letter https://doi.org/10.7554/eLife.25727.027
Author response https://doi.org/10.7554/eLife.25727.028

# Additional files

## Supplementary files

• Source code 1. Scripts used for analysis of behavioral data. *sort_fish_sttime_192.pl* is a Perl script (*Prober et al., 2006*) that converts data acquired by the Viewpoint videotracker system to a format that is useful for analysis using Matlab and removes notations that are not relevant to behavioral analysis. *perl_batch_192well.m* is a Matlab script that allows the *sort_fish_sttime_192.pl*script to run on the Matlab platform. *TapAnalysis.m* is a Matlab script that analyzes tapping assay data and generates a table that lists the number of larvae that moved during each tapping event (*Singh et al., 2015*). *VT_analysis.m* is a Matlab script (modified from (*Prober et al., 2006*)) that analyzes locomotion data collected by the Viewpoint videotracker system to quantify several metrics, including activity, waking activity, sleep, sleep architecture and sleep latency. Detailed instructions on the use of these scripts will be provided upon request.
DOI: https://doi.org/10.7554/eLife.25727.022

• Transparent reporting form
DOI: https://doi.org/10.7554/eLife.25727.023

## Major datasets

The following previously published dataset was used:

| Author(s) | Year | Dataset title | Dataset URL | Database, license, and accessibility information |
|---|---|---|---|---|
| Yates A, Akanni W, Amode MR, Barrell D, Billis K, Carvalho-Silva D, Cummins C, Clapham P, Fitzgerald S, Gil L, Girón CG, Gordon L, Hourlier T, Hunt SE, Janacek SH, Johnson N, Juettemann T, Keenan S, Lavidas I, Martin FJ, Maurel T, McLaren W, Murphy DN, Nag R, Nuhn M, Parker A, Patricio M, Pignatelli M, Rahtz M, Riat HS, Sheppard D, Taylor K, Thormann A, Vullo A, Wilder SP, Zadissa A, Birney E, Harrow J, Muffato M, Perry E, Ruffier M, Spudich G, Trevanion SJ, Cunningham F, Aken BL, Zerbino DR, Flicek P | 2016 | Ensembl 2016 | http://uswest.ensembl.org/Danio_rerio/Info/Index | Publicly available at Ensembl (accession no. ENSDARG00000036227) |

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
