## [Decision Letter]

Thank you for submitting your article "Genetic and neuronal regulation of sleep by neuropeptide VF" for consideration by *eLife*. Your article has been reviewed by three peer reviewers, one of whom is a member of our Board of Reviewing Editors and the evaluation has been overseen by a Senior Editor. The following individuals involved in review of your submission have agreed to reveal their identity: David Raizen (Reviewer #3).

The reviewers have discussed the reviews with one another and the Reviewing Editor has drafted this decision to help you prepare a revised submission.

Summary:

In this study, the authors examined the functional role of RFamide-related neuropeptides in zebrafish sleep. Previous studies in *C. elegans* and *Drosophila* have suggested that RFamide-related neuropeptides are involved in invertebrate sleep, but evidence in vertebrate has been lacking. In this study, the authors used several techniques to manipulate the expression of this peptide and activity of the neurons expressing this peptide. They showed that overexpression of the peptide reduces locomotor activity and increases sleep, and inhibition of its signaling has the opposite effects. Furthermore, they identified the small number of neurons expressing this peptide and showed that activating/ablating them increases/decreases sleep. Interestingly, they found that the sensory neurons in the optical tectum are regulated by the activity of the neurons expressing this peptide, indicating that the effect is not just reflected in motor activity but also sensory processing.

All reviewers agree that the study is interesting and the results should be of wide interest to researchers who study sleep and/or neuropeptides.

Essential revisions:

1) The optogenetic behavioural experiments are not convincing. Sleep is generally reduced after light exposure, it just seems less reduced in the optogenetic experiment. It seems that the blue light used in the study causes a nonspecific arousal of the fish. The conclusion that npvf neurons are sufficient to promote sleep is not justified. It is not clear why the authors have used blue light to stimulate ReaChr, which has an excitation maximum of around 600nm (Lin et al., 2013). Using longer wavelength light should allow the authors to reduce the power by more than 90% and thus they might be able to observe an increase of quiescence above the baseline in an awake animal without cause the nonspecific effects.

The final experiment on tectal activity is also confusing. Figure 7 simply show that the tectal activity, spontaneous or evoked changes across the day, but it could all be circadian rather than sleep related. It is unclear what E-G show. The red light used to activate ReaChR evokes responses in the tectal neurons presumably through the visual pathway. The initial responses are the same between ReaChR+ and ReaChR- animals (in fact higher in + animals), but the ReaChR+ responses reduce to a lower level later on. Are we looking at the effect of adaptation or is it really the effect of enhanced sleep?

One way to address the issue with both experiments is through chemical activation. The authors have used TRPV channels in fish before to activate neuronal populations by adding capsaicin. This would avoid any problems caused by light.

2) The alignments shown in Figure 1—figure supplement 1 are potentially misleading since it is not clear that FLP-13 or FLP-24 are more similar to NPVF than to other vertebrate RFa peptides. Conversely, NPVF peptides, beyond having the RFamide motif present in all members of this family, are not convincingly more similar to FLP-13 or FLP-24 peptides than to other nematode RFa peptides. FLP-15 and FLP-21 for example both have PLRFa motifs similar to fish RFRP1. There is no bioinformatics analysis described in the paper that would support the authors' argument. This should be added, at least to the methods how it was done. Maybe showing a phylogenetic tree with all RF peptides from *C. elegans* and zebrafish.

3) Zebrafish sleep is measured by motor responses, making it difficult to completely separate the effects on the motor system and on sleep. This is not a specific criticism of this study, but it deserves to be discussed. The RFamide-related neuropeptide or the neuronal manipulations have clear effects on locomotor activity. Sleep is measured from locomotor activity, although with a carefully defined threshold. Even arousal threshold is still measured by movement. So the authors should provide a detailed discussion of why the effects they observed cannot be due to a simple effect on the motor system.

4) Figure 1—figure supplement 1. There is increased sleep with FLP-13 overexpression but this does not reach statistical significance. It would be helpful to see the individual data in the form of a box and whiskers plot--are there just 1-2 outliers that contribute most of the variance? Could this high variance be a consequence of mosaicism observed in transient transgenics?

5) Figure 2. An alternative explanation for the need for 2 or more NPVF peptides to achieve maximal sleep is that all three peptides have similar biological activities but that peptide expression level is greater when over-expressing a pre-proprotein encoding >1 peptide. Two potential experiments could inform this possibility. One is to engineer preproproteins encoding three identical peptides. If it is only the levels that matter, then such transgenes would be as effective as the wt positive control. A simpler albeit less definitive experiment is to test the effect of exposure of the hs:NPVF animals to varying temperatures (or same temperature at varying durations) on sleep. A finding that the individual NPVF peptides do not cause increased sleep with stronger activations of the hs promoter would argue against a simple level of expression explanation. If these experiments are too difficult, the authors should at least discuss the alternative explanation.

6) Figure 3. Do the drugs block NPVF OE-induced sleep? They should, if they are acting as proposed.

7) The two worm peptides chosen for study in fish are both involved in stress-induced sleep in *C. elegans*. The authors should include this point in their intro or discussion. Can they comment on the role of NPVF in stress-induced sleep in fish?

---

## [Author Response]

Essential revisions:1) The optogenetic behavioural experiments are not convincing. Sleep is generally reduced after light exposure, it just seems less reduced in the optogenetic experiment. It seems that the blue light used in the study causes a nonspecific arousal of the fish. The conclusion that npvf neurons are sufficient to promote sleep is not justified. It is not clear why the authors have used blue light to stimulate ReaChr, which has an excitation maximum of around 600nm (Lin et al., 2013). Using longer wavelength light should allow the authors to reduce the power by more than 90% and thus they might be able to observe an increase of quiescence above the baseline in an awake animal without cause the nonspecific effects.

To address this comment we repeated the experiment using a red light stimulus as suggested by the reviewers. We found that red light caused a decrease in locomotor activity and an increase in sleep in transgenic animals compared to WT siblings, similar to our result using a blue light stimulus, however the effect was more robust using blue light. Indeed, the referenced Lin et al., 2013 paper shows that blue light also produces strong electrophysiological responses in the mammalian cells tested. We also tried using less intense red light as suggested by the reviewer, but even when we reduced red light intensity by 90%, we still observed an arousing effect of this light on both transgenic and WT animals. Since we observed a stronger phenotype using blue light, we have kept this data in the paper.

To further address this concern, we used an alternative approach to stimulate NPVF neurons that avoids the potentially confounding effect of light on behavior (Figure 5—figure supplement 2). We generated two new transgenic lines: *Tg(npvf:KalTA4)*, an optimized version of the transcriptional activator Gal4 that is specifically expressed in NPVF neurons; and *Tg(UAS:TRPV1-TagRFP-T)*, in which expression of the rat TRPV1 channel is regulated by UAS elements. We previously showed that addition of the TRPV1 small molecule agonist capsaicin to the water results in specific stimulation of zebrafish neurons that express the rat TRPV1 protein (Chen et al., 2016). This approach works because the zebrafish TRPV1 ortholog contains a mutation that makes it insensitive to capsaicin, and capsaicin added to the water can access the brain because the blood brain barrier has not yet fully formed at the larval stage of development used in our study. Furthermore, capsaicin has no effect on the behavior of WT animals at the concentration used (Chen et al., 2016). We generated *Tg(npvf:KalTA4); Tg(UAS:TRPV1-TagRFP-T)* animals and showed using fluorescent *in situ* hybridization with a probe specific for *c-fos* that treatment with 2 μm capsaicin results in activation of NPVF neurons and increased sleep at night, consistent with the phenotype observed using ReaChR. While the behavioral phenotype induced using optogenetic stimulation of NPVF neurons was larger than that using TRPV1, optogenetic stimulation induced more robust *c-fos* expression than TRPV1-dependent stimulation (compare Figure 5 with Figure 5—figure supplement 2), suggesting that more robust stimulation of NPVF neurons was achieved using optogenetics. The different level of stimulation might result from different transgene expression levels, acute stimulation with ReaChR versus prolonged stimulation with TRPV1, or different effects of ReaChR and TRPV1 on neuronal physiology. Stronger c-fos expression and behavioral effects for optogenetics compared to TRPV1 is also consistent with our previous results for zebrafish hypocretin neurons (Chen et al., 2016; Singh et al., 2015), which are located next to NPVF neurons in the hypothalamus. Indeed, our previous work suggests that the TRPV1 method may primarily result in depolarization of membrane potential and increased neuronal excitability without directly evoking action potentials (Chen et al., 2016). Because we previously showed that higher capsaicin levels (≥10 uM) results in ablation of TRPV1-expressing neurons, we verified that the number of NPVF neurons was unaffected by 2 μm capsaicin (Figure 5—figure supplement 2). Taken together, these data show that chemogenetic stimulation of NPVF neurons promotes sleep, consistent with both the NPVF optogenetic and overexpression data. We thank the reviewers for suggesting this experiment, as it clarifies the sleep-promoting role of NPVF neurons.

The final experiment on tectal activity is also confusing. Figure 7 simply show that the tectal activity, spontaneous or evoked changes across the day, but it could all be circadian rather than sleep related. It is unclear what E-G show. The red light used to activate ReaChR evokes responses in the tectal neurons presumably through the visual pathway. The initial responses are the same between ReaChR+ and ReaChR- animals (in fact higher in + animals), but the ReaChR+ responses reduce to a lower level later on. Are we looking at the effect of adaptation or is it really the effect of enhanced sleep?One way to address the issue with both experiments is through chemical activation. The authors have used TRPV channels in fish before to a activate neuronal populations by adding capsaicin. This would avoid any problems caused by light.

We thank the reviewers for this good idea. As suggested, we repeated the GCaMP6s imaging experiment using chemogenetic activation of NPVF neurons (Figure 7—figure supplement 2). We want to highlight two key features of this new chemogenetic experiment that we think provide strong corroborating support for the original optogenetic experiment. First, in the new experiment we record spontaneous neural activity (Figure 7—figure supplement 2), rather than visually-evoked activity as in the previous experiment (Figure 7). Second, we performed principal component and independent component analysis (PCA/ICA) in the new experiment to automatically segment all active neurons captured throughout the entire imaged brain slice. This allowed us to quantify neural activity more comprehensively than was done in Figure 7, where we only quantified neural activity in a region of the optic tectum. With these two key differences, the new chemogenetic experiment nevertheless reveals essentially the same result as the previous optogenetic experiment, that activation of NPVF neurons leads to reduced neural activity, consistent with increased sleep.

Thus, the two live imaging experiments together show that regardless of the mechanism of activation, for either visually-evoked or spontaneous neural activity, recorded in a tectal region of interest or throughout the imaged brain slice, we observe that activation of NPVF neurons lead to decreased neural activity, consistent with increased sleep, and consistent with the behavioral phenotypes described in the manuscript.

In response to the comment that the changes in neural activity shown in Figure 7 could be due to effects of the circadian clock, rather than the sleep/wake cycle, we agree that this is possible and now emphasize this possibility in the manuscript. Nevertheless, the main point is that both optogenetic and chemogenetic stimulation of NPVF neurons results in reduced neural activity in the brain.

In response to the comment that initial responses to the red light stimulus are higher in ReaChR+ animals compared to their ReaChR- siblings (Figure 7), please note that in these panels we normalized the data according to the peak signal after the red light is turned on for each animal. For ReaChR+ animals, because the red light stimulus causes an initial increase in GCaMP6s fluorescence followed by a large decrease, the peak in fluorescence occurs at light onset. In contrast, for ReaChR- animals, GCaMP6s fluorescence increases when the red light is turned on and remains high throughout the red light illumination period, and in this case the peak GCaMP6s fluorescence occurs several trials after light onset. Thus, the apparent higher initial response in ReaChR+ animals is due to normalization of the data and does not mean that ReaChR+ animals have a stronger initial response to red light than ReaChR- animals.

2) The alignments shown in Figure 1—figure supplement 1 are potentially misleading since it is not clear that FLP-13 or FLP-24 are more similar to NPVF than to other vertebrate RFa peptides. Conversely, NPVF peptides, beyond having the RFamide motif present in all members of this family, are not convincingly more similar to FLP-13 or FLP-24 peptides than to other nematode RFa peptides. FLP-15 and FLP-21 for example both have PLRFa motifs similar to fish RFRP1. There is no bioinformatics analysis described in the paper that would support the authors' argument. This should be added, at least to the methods how it was done. Maybe showing a phylogenetic tree with all RF peptides from C. elegans and zebrafish.

Because RFamide peptide sequences are short and there is low amino acid conservation across species in the non-RFamide peptide domains of these proteins (Nassel and Wegener, 2011; Yun et al., 2014), we found that phylogenetic analysis is not informative; all zebrafish RFamide-containing proteins form a cluster that is separate from a cluster of all *C. elegans* RFamide-containing proteins, and the RFamide peptides themselves are too short for meaningful phylogenetic analysis. However, we did not intend to claim that zebrafish RFRP peptides are specific orthologs of *C. elegans*FLP-13 and/or FLP-24. Rather, our intent was to draw inferences upon the potential importance of RFamide peptides (for example FLP-13 and FLP-24) in sleep regulation in both invertebrates and vertebrates. We have softened our claims in response to this comment and to new data (see response to point 4).

3) Zebrafish sleep is measured by motor responses, making it difficult to completely separate the effects on the motor system and on sleep. This is not a specific criticism of this study, but it deserves to be discussed. The RFamide-related neuropeptide or the neuronal manipulations have clear effects on locomotor activity. Sleep is measured from locomotor activity, although with a carefully defined threshold. Even arousal threshold is still measured by movement. So the authors should provide a detailed discussion of why the effects they observed cannot be due to a simple effect on the motor system.

We acknowledge the validity of this comment . However, we suggest that a detailed discussion of this issue would be more appropriately dealt with in a review article, as we and others have done, which we have referenced in the manuscript (e.g. Allada and Siegel, 2008; Cirelli, 2009; Joiner, 2016; Oikonomou and Prober, 2017; Sehgal and Mignot, 2011; Trojanowski and Raizen, 2016).

We cannot rule out the possibility that NPVF simply affects the motor system and not specifically sleep. However, as the reviewers note, the behavioral definition of sleep requires that locomotor quiescence be associated with an increased arousal threshold, which distinguishes quiet wakefulness from sleep. We and others have shown that one or more minutes of inactivity is associated with a dramatic increase in arousal threshold in zebrafish larvae (Elbaz et al., 2012; Prober et al., 2006). This, together with the demonstration of sleep homeostasis in zebrafish (Yokogawa et al., 2012; Zhdanova et al., 2001), meets the behavioral definition of sleep. In this manuscript, we show using both gain- and loss-of-function experiments that effects of NPVF signaling on locomotor behavior are associated with changes in arousal threshold that are consistent with the behavioral definition of sleep. We also note that we and others have tested in zebrafish many genes and drugs known to regulate mammalian sleep based on EEG recordings in rodents and humans, and found that the vast majority of these genes and drugs produce the expected effects on zebrafish behavior (Chen et al., 2016; Chen et al., 2017; Prober et al., 2006; Rihel et al., 2010; Zhdanova et al., 2001). Thus, even if zebrafish sleep does not perfectly recapitulate all aspects of mammalian sleep (as it surely does not), we suggest that zebrafish studies are nevertheless useful to identify genes and pathways that will likely be relevant to mammalian sleep.

While we do not use patterns of brain activity (EEG) to define sleep and wake, as is commonly done in mammals, but rather use behavioral criteria, we note that EEGs are only a correlate of sleep, and it is unlikely that EEG patterns define a universal or even essential feature of sleep (see (Allada and Siegel, 2008) for a detailed discussion). Indeed, sleep-like EEG patterns can be observed in awake mammals (Qiu et al., 2015; Vyazovskiy et al., 2011), and non-sleep-like EEG patterns can be observed in sleeping rodents (Bergmann et al., 1987). We acknowledge that the sleep behavior observed in zebrafish is not identical to mammalian sleep. However, considering that sleep is an essential process that is conserved throughout evolution, we see no reason that sleep should be defined according to its EEG phenotype in mammals. In fact, one could argue that sleep should be defined according to its properties in the simplest organisms that have been shown to exhibit this behavior (e.g. *C. elegans, Drosophila* and recently shown in jellyfish (Hendricks et al., 2000; Nath et al., 2017; Raizen et al., 2008; Shaw et al., 2000)). Indeed, fly and nematode papers commonly use the term “sleep” to describe this behavior (for example, (Cirelli et al., 2005; Donlea et al., 2011; Koh et al., 2008; Liu et al., 2014; Shang et al., 2013; Stavropoulos and Young, 2011)), including several papers published in *eLife* (Aso et al., 2014; Beckwith et al., 2017; Haynes et al., 2015; Iannacone et al., 2017; Kayser et al., 2015; Machado et al., 2017; Schwarz and Bringmann, 2017; Shi et al., 2014; Turek et al., 2016). It would seem odd for the literature, even in the same journal, to refer to this behavioral state as “sleep” in invertebrates, but not in fish, particularly because the neuronal substrates that are thought to regulate mammalian sleep are largely conserved in fish (Kaslin et al., 2004; Kaslin and Panula, 2001; Ma, 1994, b, 1997; McLean and Fetcho, 2004; Prober et al., 2006; Sundvik et al., 2011; Yokogawa et al., 2012), but not in invertebrates.

4) Figure 1—figure supplement 1. There is increased sleep with FLP-13 overexpression but this does not reach statistical significance. It would be helpful to see the individual data in the form of a box and whiskers plot--are there just 1-2 outliers that contribute most of the variance? Could this high variance be a consequence of mosaicism observed in transient transgenics?

To circumvent potential issues of mosaicism observed in transient transgenics, we generated stable transgenic lines and re-tested behavior in the F1 generation (Figure 1—figure supplement 1). This approach results in more robust and consistent gene overexpression (Chiu et al., 2016), and allows many more animals to be tested. We found that overexpression of FLP-13 resulted in reduced locomotor activity, but had no effect on sleep, compared to WT siblings. In contrast with our data using transient transgenics, we did not observe any significant effect of FLP-24 overexpression on sleep or locomotor activity. It is perhaps not surprising that overexpression of FLP-13 but not FLP-24 induced a behavioral phenotype in zebrafish, since the mature peptides produced by FLP-13 (P[L/F]IRF) are more similar than that produced by FLP-24 (MMVRF) to the zebrafish RFRP peptides (LP[L/Q]RF). We suggest that the FLP-13 result is consistent with a conserved functional role for RFamide peptides in behavior, even though we only observe a significant effect on locomotor activity. In light of these new results we have removed FLP-24 from the manuscript and we have softened our statements regarding FLP-13.

5) Figure 2. An alternative explanation for the need for 2 or more NPVF peptides to achieve maximal sleep is that all three peptides have similar biological activities but that peptide expression level is greater when over-expressing a pre-proprotein encoding >1 peptide. Two potential experiments could inform this possibility. One is to engineer preproproteins encoding three identical peptides. If it is only the levels that matter, then such transgenes would be as effective as the wt positive control. A simpler albeit less definitive experiment is to test the effect of exposure of the hs:NPVF animals to varying temperatures (or same temperature at varying durations) on sleep. A finding that the individual NPVF peptides do not cause increased sleep with stronger activations of the hs promoter would argue against a simple level of expression explanation. If these experiments are too difficult, the authors should at least discuss the alternative explanation.

This is an excellent point. However, the fact that RFRP2 overexpression in *Tg(hs:RFRP2)/+* animals promotes sleep, in contrast to RFRP1 or RFRP3 overexpression in *Tg(hs:RFRP1)/+* or *Tg(hs:RFRP3)/+* animals (Figure 2), suggests that the different RFRP peptides have different biological activities (note that in all experiments in the paper, we compare animals that are heterozygous for a heat-shock transgene to WT siblings, so animals either have 0 or 1 copy of the transgene in their genome. We have added this note to the manuscript). We nevertheless acknowledge that the data in the original submission showing that overexpression of either RFRP1 or RFRP3 does not promote sleep, whereas overexpression of both RFRP1 and RFRP3 does, could result from a dose-dependent effect of overexpressing either 1 or 2 RFRP peptides. To test this hypothesis, we performed a simpler experiment than that suggested by the reviewers that did not require generating new transgenic lines. We compared animals that were heterozygous or homozygous for the *Tg(hs:RFRP1)* or *Tg(hs:RFRP3)* transgene to WT relatives, and thus compared behavior of animals that had 0, 1 or 2 copies of each transgene in their genome. We observed no locomotor activity or sleep phenotype in animals homozygous for the *Tg(hs:RFRP1)* transgene compared to heterozygous transgenic and WT relatives (Figure 2—figure supplement 1). We obtained similar results for RFRP3 (Figure 2—figure supplement 1). While there was a small decrease in locomotor activity for *Tg(hs:RFRP3)* homozygous animals compared to WT relatives (Figure 2—figure supplement 1), this phenotype was much smaller than that of *Tg(hs:RFRP1)/+;Tg(hs:RFRP3)/+* double heterozygous animals (Figure 2). We also observed no significant difference in the phenotypes of animals homozygous or heterozygous for the *Tg(hs:RFRP1-3)* transgene (Figure 2—figure supplement 1). Taken together, these results support the interpretation that RFRP peptides act synergistically to promote sleep, rather than acting in a dose-dependent manner.

6) Figure 3. Do the drugs block NPVF OE-induced sleep? They should, if they are acting as proposed.

We have performed additional experiments to address this comment and found that they do. RFRP1 and RFRP3 have been shown to bind NPFFR1 and NPFFR2 in mammalian cell culture, but RFRP2 does not bind to these receptors and its receptor(s) remains unknown (Hinuma et al., 2000; Liu et al., 2001). RF9 and GJ-14 have been shown to inhibit NPFFR1 and NPFFR2 in mammals (Kim et al., 2015; Simonin et al., 2006). Accordingly, we expected that RF9 and GJ-14 should block the effect of RFRP1,3 overexpression. We found that GJ-14 completely blocked RFRP1,3 overexpression-induced sleep (Figure 3—figure supplement 2), and that RF9 significantly suppressed RFRP1,3 overexpression-induced sleep as well (Figure 3—figure supplement 2).

7) Text. The two worm peptides chosen for study in fish are both involved in stress-induced sleep in C. elegans. The authors should include this point in their intro or discussion. Can they comment on the role of NPVF in stress-induced sleep in fish?

We have added text to the Discussion section addressing these points.